# RepoGraph: Enhancing AI Software Engineering with Repository-level Code Graph

**Siru Ouyang[1]\*, Wenhao Yu[2], Kaixin Ma[2], Zilin Xiao[3], Zhihan Zhang[4], Mengzhao Jia[4], Jiawei Han[1], Hongming Zhang[2], Dong Yu[2]**

[1] University of Illinois Urbana-Champaign, [2] Tencent AI Seattle Lab
[3] Rice University, [4] University of Notre Dame
siruo2@illinois.edu

## Abstract

Large Language Models (LLMs) excel in code generation yet struggle with modern AI software engineering tasks. Unlike traditional function-level or file-level coding tasks, AI software engineering requires not only basic coding proficiency but also advanced skills in managing and interacting with code repositories. However, existing methods often overlook the need for repository-level code understanding, which is crucial for accurately grasping the broader context and developing effective solutions. On this basis, we present RepoGraph, a plug-in module that manages a repository-level structure for modern AI software engineering solutions. RepoGraph offers the desired guidance and serves as a repository-wide navigation for AI software engineers. We evaluate RepoGraph on the SWE-bench by plugging it into four different methods of two lines of approaches, where RepoGraph substantially boosts the performance of all systems, leading to *a new state-of-the-art* among open-source frameworks. Our analyses also demonstrate the extensibility and flexibility of RepoGraph by testing on another repo-level coding benchmark, CrossCodeEval. Our code is available at https://github.com/ozyyshr/RepoGraph

## 1 Introduction

Recent advancements in large language models (LLMs) have showcased their powerful capabilities across various natural language processing tasks (OpenAI, 2023; Anil et al., 2023; Dubey et al., 2024), and now, coding-specific LLMs are emerging to tackle complex software engineering challenges (Hou et al., 2023; Fan et al., 2023), such as Code-Llama (Rozière et al., 2023) and StarCoder (Li et al., 2023a). These coding-specific LLMs are capable of assisting users with various software engineering tasks, even achieving human-level performance in many function-level coding tasks, such as program synthesis (Chen et al., 2021; Austin et al., 2021), code annotation (Yao et al., 2019), bug fixing (Tufano et al., 2019), and code translation (Rozière et al., 2020).

Real-world software engineering often extends beyond single function or self-contained code files. Applications are typically built as repositories containing multiple interdependent files, modules, and libraries (Bairi et al., 2024). These complex structures require a holistic understanding of the entire codebase to perform tasks such as code completion (Shrivastava et al., 2023; Ding et al., 2023), feature addition (Liang et al., 2024), or issue resolving (Jimenez et al., 2024). Recent benchmarks like SWE-Bench (Jimenez et al., 2024) have been proposed to evaluate LLMs on real-world GitHub issues. It requires LLMs to modify the repository to resolve the issue, either by fixing a bug or introducing a new feature. This task is particularly challenging because it requires navigating complex code bases, understanding intricate dependencies between code files, and ensuring that changes integrate seamlessly without introducing new issues, which highlights the difficulties in scaling from function-level to repository-level understanding, as expounded in Figure 1.

A key step in addressing repository-level tasks is to understand the structure of a repository and identify related code. To achieve this, retrieval-augmented generation (RAG) and its variants (Xiao et al.,

---
*Work done during internship at Tencent AI Seattle Lab.

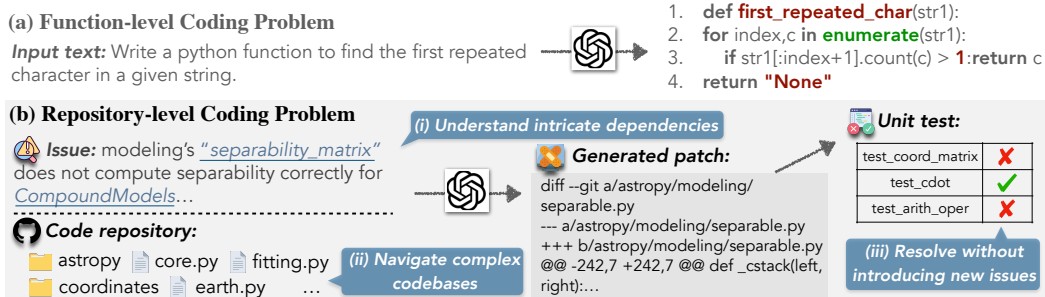

Figure 1: The illustration of *(a) a function-level coding problem* from HumanEval (Chen et al., 2021) and *(b) a repository-level coding problem* from SWE-Bench (Jimenez et al., 2024).

2023; Zhang et al., 2023; Phan et al., 2024; Wu et al., 2024) have been leveraged, in a procedural manner, to retrieve relevant code files across the repository first, providing context for LLMs for further edition. However, indexing at file-level can only identify semantically similar but not genuinely related code snippets. Instead of using RAG, recent approaches like Agentless (Xia et al., 2024) construct a skeletal format for each file, and directly prompt LLMs to identify relevant files and code lines. However, this method still treats code repositories as flat documents (Zhang et al., 2024), which suffers from limitations of repository structure such as the intricate inter-dependencies across files. An alternative approach is to design agent frameworks (Yang et al., 2024; Wang et al., 2024), which enables LLMs to interact with repositories using actions. While LLM agents can freely determine the next action based on current observations, without the grasp of global repository structures, they tend to focus narrowly on specific files, resulting in local optimums. Addressing these limitations requires going beyond semantic matching and developing techniques that enable a deeper understanding of the codebase structure. This will allow LLMs to leverage fine-grained context across multiple files and function calls, facilitating more informed, repository-wide decision-making for coding tasks.

Motivated by this, we propose REPOGRAPH, a *plug-in* module designed to help LLMs-based AI programmers leverage the code structure of *an entire repository*. REPOGRAPH is a graph structure and operates at the line level, offering a more fine-grained approach compared to previous file-level browsing methods. Each node in the graph represents a line of code, and edges represent the dependencies of code definitions and references. REPOGRAPH is constructed via code line parsing and encodes the structured representation of the entire repository. Sub-graph retrieval algorithms are then used to extract ego-graphs, which represent the relationships of a central node (in our case, specific keywords). These ego-graphs can be smoothly integrated with both procedural and agent frameworks, offering key clues that provide a more comprehensive context for LLMs to solve real-world software engineering problems.

To assess REPOGRAPH's effectiveness and versatility as a plug-in module, we integrate it with four existing software engineering frameworks and evaluate its performance using SWE-bench, a recent benchmark for AI software engineering. Experiment results show that REPOGRAPH boosts the success rate of existing methods for both agent and procedural frameworks by achieving an average relative improvement of 32.8%. We also test REPOGRAPH on CrossCodeEval to verify its transferability to general coding tasks that require repository-level code understanding. Additionally, we systematically analyze different sub-graph retrieval algorithms and integration methods. Together with error analyses, we hope to shed light on future works targeting modern AI software engineering.

## 2 RELATED WORKS

### 2.1 LLM-BASED METHODS FOR AI SOFTWARE ENGINEERING

Recently, there has been a significant increase in research focused on AI-driven software engineering, which can be broadly categorized into two primary approaches: (i) LLM agent-based frameworks and (ii) SWE-featured procedural frameworks. While this field has advanced rapidly, with most methods being released as proprietary solutions for industry applications (Cognition, 2024), our related work section will concentrate specifically on open-source frameworks.

**LLM agent-based framework** equips large language models (LLMs) with a set of predefined tools, allowing agents to iteratively and autonomously perform actions, observe feedback, and plan future steps (Yang et al., 2024; Zhang et al., 2024; Wang et al., 2024; Cognition, 2024; Ouyang et al., 2024; Tang et al., 2025). While the exact set of tools may vary across different agent frameworks, they typically include capabilities such as opening, writing, or creating files, searching for code lines, running tests, and executing shell commands. To solve a problem, agent-based approaches involve multiple actions, with each subsequent turn depending on the actions taken in previous ones and the feedback received from the environment. For example, SWE-agent (Yang et al., 2024) facilitates interactions with the execution environment by designing a special agent-computer interface (ACI). There are various actions, including "search and navigation", "file viewer and editor", and "context management". Another work, AutoCodeRover (Zhang et al., 2024), further offers fine-grained searching methods for LLM agents in better contexts without an execution process. Specifically, it supports class and function-level code search. OpenDevin (Wang et al., 2024), initiated after Devin (Cognition, 2024), is a community-driven platform that integrates widely used agent systems and benchmarks. The action space design of OpenDevin is highly flexible, requiring LLM agents to generate code on the fly.

**SWE-featured procedural frameworks** typically follow a two-step *Localize/Search-Edit* approach, as seen in existing literature (Zhang et al., 2023; Wu et al., 2024; Liang et al., 2024; Xia et al., 2024). The *localize* step focuses on identifying relevant code snippets, while the *edit* step involves completing or revising the code. Some works introduce additional steps to further enhance performance, such as the *Search-Expand-Edit* approach (Phan et al., 2024). Retrieval (Lewis et al., 2020) is a popular technique used for localization, allowing models to search for relevant code snippets from large repositories by treating issue descriptions as queries and code snippets as indexed data. Some approaches use a sliding window to ensure completeness (Zhang et al., 2023). Besides, Agentless (Xia et al., 2024) is a recently developed method that uses LLMs to directly identify relevant elements for editing within code repositories. It first recursively traverses the repository structure to generate a format that aligns files and folders vertically, with indents for sub-directories. This structure and the issue description are then input into the LLM, which performs a hierarchical search to identify the top-ranked suspicious files requiring further inspection or modification.

## 2.2 REPOSITORY-LEVEL CODING CAPABILITY

The evaluation of coding capabilities in AI systems has traditionally focused on function-level or line-level assessments (Lu et al., 2021; Chen et al., 2021; Austin et al., 2021), where individual code snippets or isolated functions are the primary units of analysis. Unlike previous studies, SWE-bench (Jimenez et al., 2024) highlights the trend of repository-level coding, driven by recent advances of coding-specific LLMs (Guo et al., 2024; Li et al., 2023b). It reflects the growing user demand to understand and contribute to entire projects rather than isolated functions (Ouyang et al., 2023), as well as solving real-world problems in an end-to-end and automatic manner.

Table 1: Comparison between our approach RE-POGRAPH and existing methods for representing the repository on various aspects. **\*RepoUnder-stander (Ma et al., 2024) and CodexGraph (Liu et al., 2024) are concurrent works to ours.**

| Model | Line-level | File-level | Repo-level |
|---|---|---|---|
| DraCo | ✗ | ✓ | ✗ |
| Aider | ✓ | ✗ | ✗ |
| RepoUnderstander* | ✗ | ✓ | ✓ |
| CodexGraph* | ✗ | ✓ | ✓ |
| REPOGRAPH | ✓ | ✓ | ✓ |

Long before the LLM era, the software engineering community has been studying the navigation of code repositories, including using eye-tracking data (Busjahn et al., 2015) and exploring interconnections (Begel et al., 2010). Then pre-trained code LLMs incorporate repository-level information such as file dependencies. But tasks at the repository level often involve more intricate call relationships within their context. Recent works like RepoCoder (Zhang et al., 2023) and RepoFuse (Liang et al., 2024) have started integrating RAG modules to harness additional information from repositories. Building on this, subsequent research has focused on embedding repository-level context into their methodologies. For instance, DraCo (Cheng et al., 2024) introduces importing relationships between files, CodePlan maintains the code changes by LLMs with incremental dependency analysis, while Aider (Gauthier, 2024) employs PageRank (Page, 1999) to identify the most significant contextual elements. RepoUnderstander (Ma et al., 2024) and CodexGraph (Liu et al., 2024) model

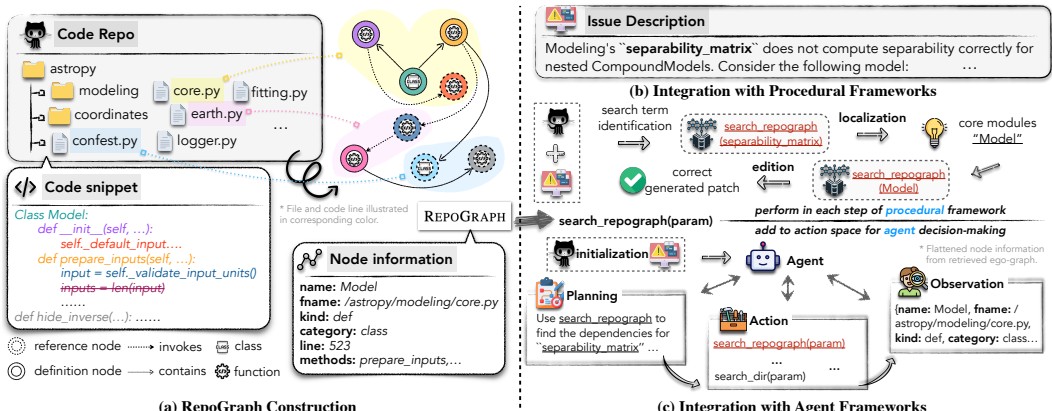

Figure 2: An in-depth illustration of *(a) the construction*, *(b) the integration with procedural frameworks*, and *(c) the integration with agent frameworks* of REPOGRAPH. Given a code repository, we first utilize AST to construct $\mathcal{G} = \{\mathcal{V}, \mathcal{E}\}$, where $\mathcal{G}$ consists of *"reference"* and *"definition"* node, $\mathcal{E}$ includes *"invoke"* and *"contain"* relations (files and code lines shown in corresponding color). The constructed REPOGRAPH are then used in procedural frameworks by adding sub-retrieval results into each step, and agent frameworks by adding graph retrieval as an additional action "search_repograph". A simplified version can be found in Figure 10.

code files as a knowledge graph. Despite similarities in representation, methods vary in how they retrieve information from these structures and utilize it for downstream tasks. Table 1 summarizes the differences between these methods and REPOGRAPH. REPOGRAPH surpasses previous approaches by effectively integrating context at the line, file, and repository levels.

# 3 REPOGRAPH

This section introduces REPOGRAPH, a novel plug-in module that can be seamlessly integrated into existing research workflows for both agent-based and procedural frameworks. The primary goal of REPOGRAPH is to provide a structured way to analyze and interact with complex codebases, enabling detailed tracing of code dependencies, execution flow, and structural relationships across the repository. In the following sections, we will provide a detailed description of REPOGRAPH's construction, its underlying representation, and its utility across various scenarios. The overall architecture is depicted in Figure 2, highlighting its key components and operational flow.

## 3.1 CONSTRUCTION

Given a repository-level coding task, the first step is to carefully examine the repository structure so that the necessary information can be collected. The input for REPOGRAPH construction is a repository, i.e., a collection of its folders and files, while the output is a structured graph, where each node is a code line, and each edge represents the dependencies in between. REPOGRAPH enables tracing back to the root cause of the current issue and gathering dependent code context to help solve the problem. The construction process of REPOGRAPH could be divided into three key steps.

**Step 1: Code line parsing.** We first traverse the entire repository using a top-down approach to identify all code files as candidates for next-step parsing. This is accomplished by filtering based on file extensions, retaining only those with relevant code file suffixes (e.g., .py) while excluding other file types (e.g., .git or requirements.txt), which might be noisy and irrelevant for coding tasks. For each code file, we utilize tree-sitter [1] to parse the code, leveraging its Abstract Syntax Tree (AST) framework. The AST provides a tree-based representation of the abstract syntactic structure of the source code, enabling the identification of key elements such as functions, classes, variables, types, and other definitions. While recognizing these definitions is crucial, tracing their usage and references throughout the code is equally important. Tree-sitter facilitates this by capturing the definitions

---

[1] https://pypi.org/project/tree-sitter-languages/

and tracking where they are utilized or referenced within the codebase. For example, in figure 2, we not only identify definitions like class Model and its inherent methods but also references like self._validate_input_units(). After processing each line of code with a tree-sitter, we selectively retain lines that involve function calls and dependency relations, discarding extraneous information. Our focus is primarily on the *functions* and *classes*, as these represent the core structural components of the code. By concentrating on these elements and their interrelationships, REPOGRAPH optimizes the analysis process by excluding less significant details, such as individual variables, which tend to be redundant and less relevant for further processing.

**Step 2: Project-dependent relation filtering.** After the previous parsing step, we obtain code lines with calling and dependency relations. However, not all relations are useful for fixing issues. Specifically, many default and built-in function/class calls could distract from the project-related ones. Therefore, we additionally introduce a filtering process that excludes the repository-independent relations. Two types of such relations exist: (i) *global relation* refers to Python standard and built-in functions and classes. (ii) *local relation* are introduced by third-party libraries, which are specific to the current code file. For global relations, we maintain a comprehensive list of methods from standard and built-in libraries, excluding any identified relations from this list. The list is empirically constructed by gathering methods of the builtins library and default methods such as "list" and "tuple". For example, in figure 2, line inputs = len(input) is excluded since "len" is a default method. For local relations, we parse import statements in the code to identify third-party methods that are included, and exclude them accordingly. A detailed illustration of this step can be found in Figure 11.

**Step 3: Graph organization.** At this stage, we construct REPOGRAPH using code lines as the fundamental building units. The graph can be represented as $\mathcal{G} = \{\mathcal{V}, \mathcal{E}\}$, where $\mathcal{V}$ represents the set of nodes, with each node corresponding to a line of code, and $\mathcal{E}$ represents the set of edges, capturing the relationships between these code lines. Each node in $\mathcal{V}$ contains attributes to represent its meta-information, such as line_number, file_name, directory, etc. Additionally, we classify each code line as either a "definition" (def) or a "reference" (ref) to a particular module. A "def" node corresponds to the line where a function, class, or variable is initially defined, while a "ref" node indicates a code line where this entity is referenced or invoked elsewhere in the code. Similar to soft links to "def" nodes, "ref" nodes also represent other variations of invoking methods. For example, in Figure 2, the class definition *"class Model"* would be a "def" node, while any subsequent usages of Model would be "ref" nodes. Each "def" node may have multiple "ref" nodes associated with it, as a single function or class can be referenced in various places throughout the code. We define two types of edges: $\mathcal{E}_{invoke}$ and $\mathcal{E}_{contain}$. The triple $(\mathcal{V}1, \mathcal{E}_{contain}, \mathcal{V}_2)$ denotes that $\mathcal{V}_1$ (e.g., a function definition) contains another module $\mathcal{V}_2$ (e.g., an internal function or class). The edge $\mathcal{E}_{contain}$ typically connects a "def" node to its internal components. In contrast, $\mathcal{E}_{invoke}$ represents an invocation relationship, usually connecting a "def" node to a "ref" node, where the reference node includes a dependency on the definition.

## 3.2 UTILITY

The constructed REPOGRAPH serves as a structured representation of the current repository and facilitates better-related information collection and aggregation. For information collection based on REPOGRAPH, specifically, we use one search term each time for subgraph retrieval. Search terms are the key functions or classes that are determined by current states. For example, "separability_matrix" is the initial search term in Figure 2(c). We retrieve the $k$-hop ego-graphs (Hu et al., 2024) with the search term in the centric. The ego-graph is crucial for solving the problem because it focuses on the immediate relationships (Jin et al., 2024) around the search term, capturing the relevant dependencies and interactions within the repository, which is key to understanding the functional context. Additionally, the retrieved content explicitly contains information at both the method and line levels and implicitly expresses the grouping at the file level.

This process is abstracted via *search_repograph()* as illustrated in the middle of Figure 2. The retrieved $k$-hop ego-graph will be flattened for further processing. We also tried other variants for integration later in Section 5.2 and their performance in Table 4. We narrate how REPOGRAPH could be plugged in with existing representative research lines in the following.

**Integration with procedural framework.** In a procedural framework, LLMs are usually prompted in "localization" and "editing" stages with the given repository context and issue description. In this case, we use *search_repograph()* before both stages, leveraging our REPOGRAPH to assist in making more informed decisions at each step. For example, in Figure 2, we first include the subgraph of *"separability_matrix"* for localization, and then use the localized result *"Model"* to search in the edition stage. To implement the strategy, we flatten the context of retrieved ego-graphs and append it as part of the prompt. As a result, the LLM generation is conditioned on both retrieved ego-graphs and the context provided by baseline methods, helping the model preserve nuances.

**Integration with agent framework.** A significant difference in existing agent frameworks is the action space design, as expounded in Section 2. To leverage the power of REPOGRAPH, we put *search_repograph()* as an additional action in the action space. The agent decides when and where to use this action. The search term is also determined by the agent accordingly. The returned subgraph will be flattened and used as an observation for the next state.

## 4 EXPERIMENTS

### 4.1 SETUP

We evaluated REPOGRAPH as a plug-in component, i.e., integrated into existing baseline models of the two aforementioned research lines to assess its performance. We use the same baseline settings and configurations when incorporating REPOGRAPH to ensure a fair comparison.

**Dataset.** We test REPOGRAPH in SWE-bench-Lite[2]. Each problem in the dataset requires submitting a patch to solve the underlying issue described in the input issue description. The goal is to generate a patch that accurately revises the relevant portions of the codebase within the repository, ensuring that all test scripts included in the dataset are successfully executed.

**Baselines.** We integrate REPOGRAPH with representative methods from both aforementioned research lines. (i) For procedural frameworks, we evaluate the widely used traditional method, RAG (Lewis et al., 2020), as well as Agentless (Xia et al., 2024), an open-source state-of-the-art approach in this direction. For RAG, we follow its initial setting and use BM25 for file-level retrieval. After that, we append the context of REPOGRAPH after the code files as part of the prompt. Agentless first performs a hierarchical localization in terms of "file-class/function-edits" and then conducts repair based on localization. The context of REPOGRAPH is inserted in every step of Agentless. (ii) For agent frameworks, we consider SWE-agent (Yang et al., 2024) and AutoCodeRover (Zhang et al., 2024). For both frameworks, we add an additional action "search_repograph" for the LLM agent as described in Section 3. All the choices in the two research lines incorporate GPT-4 and GPT-4o-based methods to ensure generalization. Detailed implementations and prompts used can be found in Appendix A.

**Evaluation metrics.** We evaluate all methods across two key dimensions: Accuracy and Average Cost. (i) For accuracy, we report the *resolve rate* and *patch application rate*. The resolve rate represents the percentage of issues successfully resolved across all data points. An issue is considered resolved if the submitted patch passes all test scripts. To assess the patch application rate, we attempt to apply the generated patches to the repository using the `patch` program, counting only successful applications toward this metric. (ii) To evaluate cost efficiency, we report two metrics: *average cost* and *average tokens*, which refer to the inference cost and the number of input/output tokens used when querying the LLMs, respectively.

**Configurations.** We use the same GPT version as in the baselines in the experiments. We used GPT-4o (`2024-05-13`) and GPT-4-Turbo (`gpt-4-1106-preview`) from OpenAI for evaluation and analyses in our experiments. All evaluation processes are performed in a containerized Docker environment [3], ensuring stability and reproducibility, made possible through contributions from the open-source community. Plug-ins with procedural frameworks usually take around 2-3 hours to finish. For agent frameworks, the inference time is larger, up to around 10 hours.

---

[2]Current leaderboard could be found at https://www.swebench.com/
[3]https://github.com/aorwall/SWE-bench-docker
[5]https://github.com/swe-bench/experiments/tree/main/evaluation/lite

Table 2: Results of REPOGRAPH with open-source baselines in two research lines, including procedural and agent frameworks. Numbers of accuracy-related metrics are directly taken from the leaderboard, while the cost-related ones are computed from the corresponding trajectories[5].

| Methods | LLM | Accuracy | | | Avg. Cost | |
|---|---|---|---|---|---|---|
| | | *resolve* | *# samples* | *patch apply* | *$ cost* | *# tokens* |
| ***Procedural frameworks*** | | | | | | |
| RAG | GPT-4 | 2.67 | 8 | 29.33 | $0.13 | 11,736 |
| +REPOGRAPH | GPT-4 | 5.33 ↑2.66 | 16 ↑8 | 47.67 ↑18.34 | $0.17 | 15,439 |
| Agentless | GPT-4o | 27.33 | 82 | 97.33 | $0.34 | 42,376 |
| +REPOGRAPH | GPT-4o | 29.67 ↑2.34 | 89 ↑7 | 98.00 ↑0.67 | $0.39 | 47,323 |
| ***Agent frameworks*** | | | | | | |
| AutoCodeRover | GPT-4 | 19.00 | 57 | 83.00 | $0.45 | 38,663 |
| +REPOGRAPH | GPT-4 | 21.33 ↑2.33 | 64 ↑7 | 86.67 ↑3.67 | $0.58 | 45,112 |
| SWE-agent | GPT-4o | 18.33 | 55 | 87.00 | $2.53 | 498,346 |
| +REPOGRAPH | GPT-4o | 20.33 ↑2.00 | 61 ↑6 | 90.33 ↑3.33 | $2.69 | 518,792 |

## 4.2 EXPERIMENT RESULTS

Table 2 presents the main evaluation results of all baseline methods and the corresponding performance with REPOGRAPH (+REPOGRAPH) as a plug-in in the SWE-bench-Lite test set. We also report the number of correct samples for each method. The performance increase is marked by ↑*num*. We also tested other (open-source) LLMs using Claude-3.5-Sonnet (`claude-3-5-sonnet-20240620`) (Anthropic, 2024) from Anthropic in Appendix C. Based on the results, we have the following key observations:

**(i) REPOGRAPH brings consistent performance gain for all combinations of frameworks and LLM model bases.** Specifically, REPOGRAPH achieves an absolute improvement of +2.66 and +2.34 in terms of the resolve rate for RAG and Agentless, respectively, which is 99.63% and 8.56% of relative improvement. The notable improvement demonstrates the effectiveness of our REPOGRAPH in adapting to various scenarios by inducing relevant code context and performing precise code editings. Additionally, our best performance so far by plugging in Agentless, 29.67, achieves the *state-of-the-art* performance on the benchmark[6] among all open-source methods.

**(ii) Performance gain brought by REPOGRAPH is slightly larger on procedural frameworks than agent ones.** With procedural frameworks, REPOGRAPH correctly fixes more issues than agent ones. This could be due to two primary reasons. Firstly, we observed that mature procedural frameworks tend to achieve better baseline performance than agent-based frameworks on SWE-bench. The initial definitive nature of procedural frameworks, with their well-defined running flow and structure, allows them to leverage plug-ins more effectively. Another reason is that this deterministic behavior reduces the complexity that arises from dynamic decision-making, a key characteristic of agent-based systems, thereby enabling a smoother integration of performance improvements.

**(iii) Performance gain brought by REPOGRAPH does not rely on more costs.** We also compute and report each method's average cost and token consumption. By introducing REPOGRAPH, we manage to reduce the costs associated with managing the entire repository while achieving comparable or even superior performance. As shown in Table 2, the additional token cost introduced by RepoGraph is justified given the significant performance improvement it enables, particularly when compared to the substantially higher costs incurred by previous methods like SWE-agent. This indicates REPOGRAPH 's performance gains are not mainly due to increased token usage.

**(iv) Average costs are generally larger on agent frameworks with REPOGRAPH.** This phenomenon is especially obvious with SWE-agent, as it allows the agent to freely determine the next action based on the current observation. We also found that the integration with agent frameworks usually leads to larger cost increases, as exemplified by +0.13$ and +0.18$ with AutoCodeRover and SWE-agent, respectively. The reason lies in the large exploration space in agent frameworks. The agents might call the *search_repograph()* action many times, which leads to the explosion of

---

[6] `https://www.swebench.com/`. We kindly request that reviewers refrain from intentionally checking submitter information on the leaderboard to maintain the integrity of the double-blind review process.

prompt contexts. We encourage users to be mindful of cost and to adopt a more granular approach to cost control when integrating REPOGRAPH into the agent framework in the future.

## 5 ANALYSIS

This section presents a detailed analysis to demonstrate that the additional context provided by REPOGRAPH is beneficial for the task. We begin by analyzing localization accuracy in comparison to the gold-standard patch. Next, we explore various REPOGRAPH configurations, focusing on how the additional context can be effectively integrated into the existing system. Finally, we perform an in-depth error analysis, highlighting aspects where REPOGRAPH can be further improved. For more analyses including resolve rate in various aspects and action distributions of agent frameworks, please refer to Appendices C.

### 5.1 LOCALIZATION COVERAGE

A crucial step in issue resolution is accurately identifying the correct locations within the code that require modification. Proper localization is essential, as it forms the foundation for generating an effective and accurate patch. Without this step, the quality of the fix may be compromised, leading to incomplete or incorrect solutions. We compute the percentage of problems where the edit locations match the ground truth patch in three granularity. Namely, file-level, function-level, and line-level. We report that a patch contains the correct location if it edits a *superset* of all locations in the ground truth patch.

Table 3: Percentage of problems for accurate edition localizations with respect to file, function, and line levels. All the numbers are computed from the corresponding generated patches.

| Methods | LLM | *file* | *function* | *line* |
|---|---|---|---|---|
| RAG | GPT-4 | 47.3 | 23.3 | 12.7 |
| +REPOGRAPH | GPT-4 | 51.7↑4.4 | 25.3↑2.0 | 14.3↑1.6 |
| Agentless | GPT-4o | 68.7 | 51.0 | 34.3 |
| +REPOGRAPH | GPT-4o | 74.3↑5.6 | 54.0↑3.0 | 36.7↑2.4 |
| *Agent frameworks* | | | | |
| AutoCodeRover | GPT-4 | 62.3 | 42.3 | 29.0 |
| +REPOGRAPH | GPT-4 | 69.0↑4.7 | 45.7↑3.4 | 31.7↑2.7 |
| SWE-agent | GPT-4o | 61.7 | 46.3 | 32.3 |
| +REPOGRAPH | GPT-4o | 67.3↑5.6 | 49.3↑3.0 | 35.0↑2.7 |

Table 3 presents the results of our analysis. We observed that integrating REPOGRAPH with all baseline methods significantly improves file-level accuracy, whereas the enhancement of accuracy in line-level is comparatively modest. This result aligns with our expectations, as file-level localization is the most coarse-grained, making it inherently easier to improve. In contrast, line-level localization, being the most fine-grained, poses a greater challenge due to its need for more precise identification of code segments. Additionally, we found that although line-level accuracy improvements are more pronounced for agent frameworks, their overall resolve rate is lower than that of procedural frameworks, as shown in Table 2. This discrepancy can be attributed to the fact that localization, while necessary for generating a final patch, is insufficient. The success of the final revision still heavily relies on the underlying capabilities of LLMs. Agent frameworks, designed to operate in a trial-and-error fashion, are particularly susceptible to error accumulation. As these frameworks iteratively refine their patches, small inaccuracies in earlier localization steps can propagate and magnify throughout the process, ultimately reducing the overall resolve rate. Procedural frameworks, on the other hand, follow a more structured and deterministic approach to localization and patch generation. They typically localize and fix issues in a single, more direct step, which can help mitigate the compounding of errors.

### 5.2 INVESTIGATION OF REPOGRAPH VARIANTS

In this section, we investigate the efficacy of various combinations of sub-graph retrieval and integration techniques as outlined in Section 3. We explore two sub-graph retrieval variants and two integration methods. Specifically, for sub-graph retrieval, we index $k$-hop ego-graphs where $k$ is set to 1 and 2. We limit our exploration to $k$ values up to 2 due to the extensive context required for integration and the potential introduction of noise or irrelevant nodes for $k \geq 3$. For the integration methods, we employ two distinct approaches: (i) directly flattening the textual sub-graph by explic-

Table 4: The number of nodes, edges, and tokens of REPOGRAPH and its variants. For different retrieval and integration variants, we report the average number on the test set. "summ." refers to the summarized version by LLMs of the retrieved ego-graph.

| Metrics | REPOGRAPH | 1-hop + flatten | 1-hop + summ. | 2-hop + flatten | 2-hop + summ. |
|---|---|---|---|---|---|
| # Nodes | 1419.3 | 11.6 | 11.6 | 54.5 | 54.5 |
| # Edges | 26392.1 | 37.1 | 37.1 | 89.9 | 89.9 |
| # tokens | - | 2310.7 | 717.5 | 10505.3 | 1229.2 |
| *resolve rate* | - | 29.67 | 28.33 | 26.00 | 28.67 |

itly detailing the relationships between the search term and its neighboring nodes, and (ii) leveraging an LLM first to summarize the sub-graph in terms of the core modules and salient dependencies, before proceeding with further processing. Detailed implementations of these variants and prompts used can be found in Appendix B.1.

We begin by presenting some statistics for REPOGRAPH and its various configurations. Performance evaluations are conducted on Agentless with REPOGRAPH integrated as a plug-in module. Table 4 reports the number of nodes and edges within the (sub)-graphs. Notably, the average number of nodes and edges for REPOGRAPH across the SWE-bench dataset is quite substantial, featuring over 1,000 nodes and 25,000 edges. This highlights the comprehensive nature of the constructed structure. For the different variants, when $k = 1$, the information within REPOGRAPH is concentrated around the search term, resulting in an average of 11.6 nodes and 37.1 edges. As $k$ increases to 2, the retrieved ego-graph expands exponentially, reaching an average of $54.5$ nodes and $89.9$ edges. Moreover, directly flattening the retrieved ego-graph often significantly increases the token count, frequently reaching several thousand tokens. However, utilizing an LLM as an additional summarizer greatly reduces the token count, typically around a thousand.

Table 4 presents the resolve rates for four variants of our method. Notably, while the 2-hop variant incorporates additional information, directly flattening this information results in the poorest performance, with a resolve rate of $26.00\%$, even lower than the original baseline. In contrast, incorporating summarization via the LLM significantly alleviates context length constraints and enhances information organization, thereby improving performance. For the 1-hop variant, however, we observed that adding summarization actually degrades performance, reducing the resolving rate to $28.33\%$. We hypothesize that this occurs because the flat 1-hop ego-graph already contains comprehensive information that fits within the LLM's context window; thus, summarization may introduce inevitable information loss.

## 5.3 TRANSFERIBILITY TEST

To demonstrate the representational power of REPOGRAPH for repositories and its transferability to tasks requiring an understanding of repository structures, we conducted experiments using the CrossCodeEval benchmark (Ding et al., 2023). CrossCodeEval is a multilingual code completion benchmark that is derived from real-world repositories. It is designed to emphasize numerous cross-file dependencies. We focus on problems using *Python* as the evaluation programming lan-

Table 5: Results on the subset of CrossCodeEval with GPT-4o and Deepseek-Coder-V2-Lite-Instruct as the backbone LLMs.

| Methods | Code Match | | Identifier Match | |
|---|---|---|---|---|
| | EM | ES | EM | F1 |
| Deepseek-Coder | 10.2 | 57.3 | 16.6 | 49.1 |
| +REPOGRAPH | 19.7 | 67.8 | 29.3 | 58.9 |
| GPT-4o | 10.5 | 59.6 | 16.8 | 47.9 |
| +REPOGRAPH | 28.7 | 68.9 | 36.0 | 61.3 |

guage, leading to 2,665 samples. For evaluation metrics, we follow the settings in the original paper and measure performance with code match and identifier match metrics, assessing accuracy with exact match (EM), edit similarity (ES), and F1 scores. In our experiments, the search terms are determined to be the function within which the current line is to be completed.

Table 5 demonstrates the results on CrossCodeEval using GPT-4o and Deepseek-Coder as the backbone language model. The original LLMs struggle with repository-level tasks, evidenced by an EM score of only $10.8$ for code matching and $16.7$ for identifier matching. These results indicate significant limitations in handling code structure and variable usage in a broader repository context.

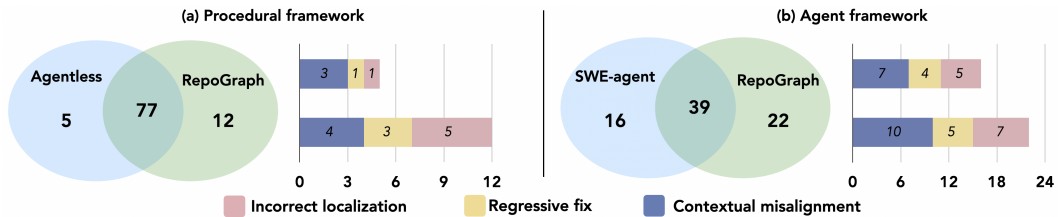

Figure 3: Venn diagram of REPOGRAPH and baselines on (a) procedural framework and (b) agent framework on SWE-Bench-Lite. We also plot the error distribution of failing cases against counterparts, e.g., detailed error distribution of 12 cases REPOGRAPH succeeds while Agentless fails.

However, with the integration of the REPOGRAPH method, there is a substantial improvement across all metrics with both LLMs. Particularly, we find that the improvement brought by REPOGRAPH is more pronounced when integrated with GPT-4o. A potential reason is that GPT-4o is better at interpreting the additional context information provided for better code reasoning. These improvements suggest that incorporating repository-level knowledge, as facilitated by REPOGRAPH, greatly enhances the model's ability to understand and generate more contextually accurate and semantically consistent code.

## 5.4 ERROR ANALYSIS

We want to compare REPOGRAPH with the corresponding baselines to see the distribution of resolved cases and analyze the error reasons for unresolved cases. We plot a Venn diagram for representative methods in both procedural and agent frameworks in Figure 3, respectively. We manually examined all the unique error cases and defined three error categories. **(i) Incorrect localization** refers to the failure in accurately identifying code snippets, **(ii) contextual misalignment** happens when the generated patch fails to align with the broader context of the codebase, and **(iii) regressive fix** introduces new issues in resolving the original issues. More examples are in Appendix D.

We found that the improvement in agent frameworks is more complementary than procedural frameworks, with larger uniquely resolved cases of 22 compared with 12 in procedural frameworks. Together, they make even larger performance of 31.33% and 22.33%. The reason could also be attributed to the determinism of procedural frameworks. As a plug-in module, REPOGRAPH tends to make modifications on existing deterministic processes, resulting in larger overlaps in resolved issues compared with baselines. Agent frameworks, on the other hand, have quite different action distributions with REPOGRAPH as a plug-in (please refer to Appendix A.2). Therefore, the uniquely resolved cases are more compared with procedural frameworks. For error distributions, contextual misalignment is the most prevalent error type, followed by incorrect localization and regressive fixes for all methods, suggesting that while localization is often correct, the applied solutions may regress or fail to integrate contextually. The phenomenon also echoes the conclusion obtained in Section 5.1. This is intuitive as all the existing methods focus on providing a more comprehensive and desired context for LLMs to solve the task, which fundamentally depends on the power of backbone LLMs. We also found that when integrated with REPOGRAPH, the proportion of all three error types decreases (specifically for "incorrect localization"), indicating that REPOGRAPH is specifically useful in aggregating the related contexts of the current to-be-fixed issue.

## 6 CONCLUSION AND DISCUSSION

This paper introduces REPOGRAPH, a plug-in module for modern AI software engineering. REPOGRAPH operates at the code-line level and offers desired navigation through code bases by incorporating and aggregating information at the line level, file level, and repository level through subgraph retrieval of ego-graphs. Extensive experiments on real-world SWE tasks show that REPOGRAPH significantly boosts the performance of both procedural and agent frameworks. In addition, REPOGRAPH is proved to be smoothly and effectively transferred to other tasks that require repository-level understanding. Future work could further investigate the generalizability of REPOGRAPH across different programming languages, frameworks, and developer environments. Real-time execution and feedback could also be explored to further enhance REPOGRAPH towards a dynamic version for better AI software engineering abilities.

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

## Contents of Appendix

## A  DETAILED IMPLEMENTATIONS FOR EACH BASELINE METHOD

This section details the implementation of all four methods in procedural and agent research lines mentioned in Section 4.1.

### A.1  PROCEDURAL FRAMEWORKS

Figure 4 and Figure 5 illustrate the instructions we used for procedural frameworks, including localization and edition, respectively.

**Instruction**

Please review the following GitHub problem description and relevant files, and provide a set of locations that need to be edited to fix the issue. You will also be given a list of function/class dependencies to help you understand how functions/classes in relevant files fit into the rest of the codebase. The locations can be specified as class names, function or method names, or exact line numbers that require modification.

**Template**

### GitHub Problem Description ###
{problem_statement}

### Related Files ###
{file_contents}

### Function/Class Dependencies ###
{repo_graph}

###
Please provide the class name, function or method name, or the exact line numbers that need to be edited.

**Demonstrations**

```
### Examples:
```
full_path1/file1.py
line: 10
class: MyClass1
line: 51
```
full_path2/file2.py
function: MyClass2.my_method
line: 12

full_path3/file3.py
function: my_function
line: 24
line: 156
```

Return just the location(s)

Figure 4: Instructions used in the procedural framework to localize to detailed files and code lines of edition.

We flattened the context of the retrieved ego-graph from REPOGRAPH into the template of the instructions. Specifically, in both "localization" and "edition" stages, REPOGRAPH is flattened in the part of Function/Class Dependencies so that the LLMs could better understand its context.

### A.2  AGENT FRAMEWORKS

We list all the instructions in every step of agent frameworks. The overall and system instruction is shown in Figure 6.

The system instruction defines the task setting and the template for each response. We add "search_repograph" as a new action for the agent to use in the command_docs, with its signature listed in Figure 7.

We also plot the frequency for action invoked for both SWE-agent and SWE-agent with REPO-GRAPH in Figure 8. We can see that with REPOGRAPH, the maximum turn of finishing the task is reduced from 38 to 35. Also, we computed the average turn to finish the task, which demonstrates a similar trend of 21.47 to 19.12, a significant improvement in efficiency while maintaining effectiveness. We also observe that the action "search_repograph" is invoked mostly in the first 15 rounds of conversation with LLMs. It is more precise than the original action of "search_dir", "search_file", and "find_file".

| | |
|---|---|
| **Instruction** | We are currently solving the following issue within our repository. Here is the issue text: {problem_statement}

Below are some code segments, each from a relevant file. One or more of these files may contain bugs. {content}

To help you better understand the contexts of the code segments, we provide a set of dependencies of the code segments. The dependencies reflect how the functions/classes in the code segments are referenced in the codebase. {dependencies} |
| **Template** | Please first localize the bug based on the issue statement, and then generate *SEARCH/REPLACE* edits to fix the issue.

Every *SEARCH/REPLACE* edit must use this format:
1. The file path
2. The start of search block: <<<<<<< SEARCH
3. A contiguous chunk of lines to search for in the existing source code
4. The dividing line: =======
5. The lines to replace into the source code
6. The end of the replace block: >>>>>>> REPLACE |
| **Demonstrations** | Here is an example:

### mathweb/flask/app.py
<<<<<<< SEARCH
from flask import Flask
=======
import math
from flask import Flask
>>>>>>> REPLACE | Please note that the *SEARCH/REPLACE* edit REQUIRES PROPER INDENTATION. If you would like to add the line ' print(x)', you must fully write that out, with all those spaces before the code!
Wrap the *SEARCH/REPLACE* edit in blocks ```python...```. |

Figure 5: Instruction used for fixing an issue based on the identified locations in certain template.

**SETTING:**
You are an autonomous programmer, and you're working directly in the command line with a special interface.
The special interface consists of a file editor that shows you {WINDOW} lines of a file at a time.
In addition to typical bash commands, you can also use the following commands to help you navigate and edit files.

**COMMANDS:**
{command_docs}
Please note that THE EDIT COMMAND REQUIRES PROPER INDENTATION.
If you'd like to add the line ' print(x)' you must fully write that out, with all those spaces before the code! Indentation is important and code that is not indented correctly will fail and require fixing before it can be run.

**RESPONSE FORMAT:**
Your shell prompt is formatted as follows:
(Open file: <path>) <cwd> $

You need to format your output using two fields; discussion and command.
Your output should always include _one_ discussion and _one_ command field EXACTLY as in the following example:

**DISCUSSION**
First I'll start by using ls to see what files are in the current directory. Then maybe we can look at some relevant files to see what they look like.
ls -a

You should only include a *SINGLE* command in the command section and then wait for a response from the shell before continuing with more discussion and commands. Everything you include in the DISCUSSION section will be saved for future reference.
If you'd like to issue two commands at once, PLEASE DO NOT DO THAT! Please instead first submit just the first command, and then after receiving a response you'll be able to issue the second command.
You're free to use any other bash commands you want (e.g. find, grep, cat, ls, cd) in addition to the special commands listed above.
However, the environment does NOT support interactive session commands (e.g. python, vim), so please do not invoke them.

Figure 6: The signature of our new action "search_repograph" for agent frameworks.

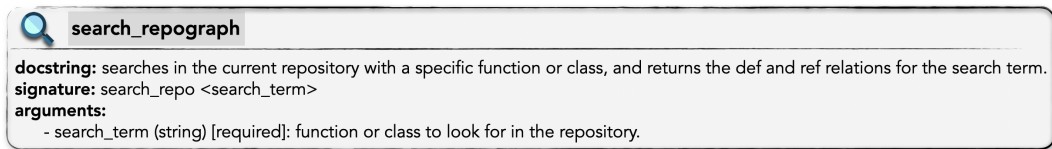

Figure 7: The signature of our new action "search_repograph" for agent frameworks.

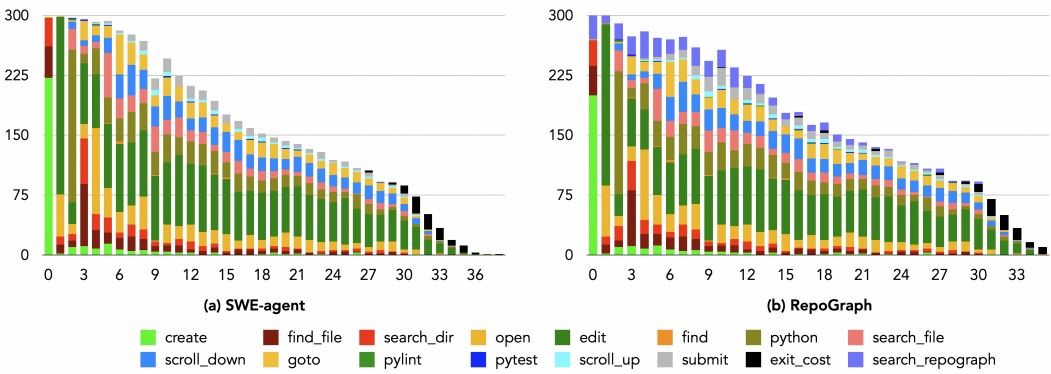

Figure 8: The frequency with which actions are invoked at each turn by *(a) SWE-agent* and *(b) SWE-agent w/*RepoGraph.

## B    Detailed implementations for RepoGraph

### B.1    RepoGraph variant

In this section, we provide the implementations for variants of RepoGraph mentioned in Section 5.2.

In addition to directly flattening the retrieved ego-graph, we propose leveraging large language models (LLMs) to first summarize the context. The full prompt, along with sample input and output, is provided in Figure 9.

### B.2    RepoGraph components

We first provide a simplified version of Figure 2 in Figure 10, which helps readers quickly catch the overall pipeline of integration with RepoGraph.

We also provide an illustration in Figure 11 to explain in detail how we exclude project-independent dependencies mentioned in Section 3.1 Step 2.

## C    Additional Results

### C.1    Results on other LLMs

We conduct additional experiments on SWE-Bench-Lite with Claude-3.5-Sonnet, and the results are shown in Table 6. We found that with Claude-3.5-Sonnet, the results of baselines are improved a little. This is intuitive as Claude-3.5-Sonnet is designed for better coding tasks. Additionally, integrating RepoGraph with the baseline methods can still boost the performance by a large margin, which further prove the effectiveness and extendability of RepoGraph on different foundational models.

**Instruction**

You will be given a list of code lines and their meta data, please summarize them in terms of the most salient function/classes and the dependencies, including invocation and inheritance.
For each instance, please keep the following output format:
** Key Classes and Functions:**
1.  [Class/Function name]:
    • [Description of the class/function]
**Core Dependencies:**
1.  [Class/Function name]:
    • [Description of the dependency]

**Input**

**location:** astropy/modeling/core.py **lines** 1765 - 1885 **name:** render
**contents:** \nclass Model(metaclass=_ModelMeta):\n   def render(self, out=None, coords=None):\n        \"\"\"\n        Evaluate a model at fixed positions, respecting the `bounding_box.\n\n        The key difference relative to evaluating the model directly is that\n       this method is limited to a bounding box if the Model.bounding_box\n        attribute is set.\n\n        Parameters\n   out : numpy.ndarray, optional\n        …

**location:** astropy/modeling/core.py **lines** 4537 - 4633 **name:** render_model
**contents:** \ndef render_model(model, arr=None, coords=None):\n   \"\"\"\n    Evaluates a model on an input array. Evaluation is limited to\n   a bounding box if the Model.bounding_box attribute is set.\n\n   Parameters\n   ----------\n   model : Model\n       Model to be evaluated.\n   arr : numpy.ndarray, optional\n        Array on which the model is evaluated.\n   coords : array-like, optional\n        Coordinate arrays mapping to arr, such that\n        arr[coords] == arr.\n\n   Returns:…

**location:** astropy/nddata/utils.py **lines** 546 - 654 **name:** __init__
**contents:** \nclass Cutout2D:\n   def __init__(\n        self, data, position, size, wcs=None, mode=\"trim\", fill_value=np.nan, copy=False\n   ):\n       if wcs is None:\n            wcs = getattr(data, \"wcs\", None)\n       if isinstance(position, SkyCoord):\n           if wcs is None:\n            raise ValueError(\"wcs must be input if position is a SkyCoord\")\n           position = skycoord_to_pixel(position, wcs, mode=\"all\")  # (x, y)\n\n       if np.isscalar(size):\n            size = np.repeat(size, 2) …

**location:** astropy/modeling/core.py **lines** 3872 - 3986 **name:** render
contents: \nclass CompoundModel(Model):\n   def render(self, out=None, coords=None):\n        \"\"\"\n        Evaluate a model at fixed positions, respecting the bounding_box.\n\n        The key difference relative to evaluating the model directly is that\n       this method is limited to a bounding box if the Model.bounding_box\n        attribute is set. …

**Output**

**Key Classes and Functions:**
1.  Model (Class):
    • **Location:** astropy/modeling/core.py (line 1765-1865)
    • **Function:** This is the base class for models in the astropy library. It contains a render method, which evaluates the model at fixed positions while respecting the bounding_box attribute. This method ensures that the evaluation occurs over a defined region, with support for various inputs like coordinates and output arrays.
    • **Invocation:** The Model class uses a custom metaclass _ModelMeta. The render method works with numpy arrays and uses helper methods like extract_array and add_array for array manipulation.
2.  Cutout2D (Class):
    • **Location:** astropy/nddata/utils.py (line 546-654)
    • **Function:** A utility class that creates a cutout from a larger 2D array (like an image) centered around a specific position. The __init__ method handles coordinate transformations, shape adjustments, and manages World Coordinate System (WCS) integration for astronomical data.
    • **Invocation:** ……
3.  render_model (Function): ……

**Core Dependencies:**
1.  extract_array:
    • Extracts a portion of a 2D or 3D array based on the provided coordinates and shape, allowing model evaluations to occur within specific sub-regions, used by Model.render, CompoundModel.render, and render_model.
2.  skycoord_to_pixel
    • Astronomical utilities for converting between celestial coordinates and pixel positions, used in Cutout2D
3.  WCS:
    • ……

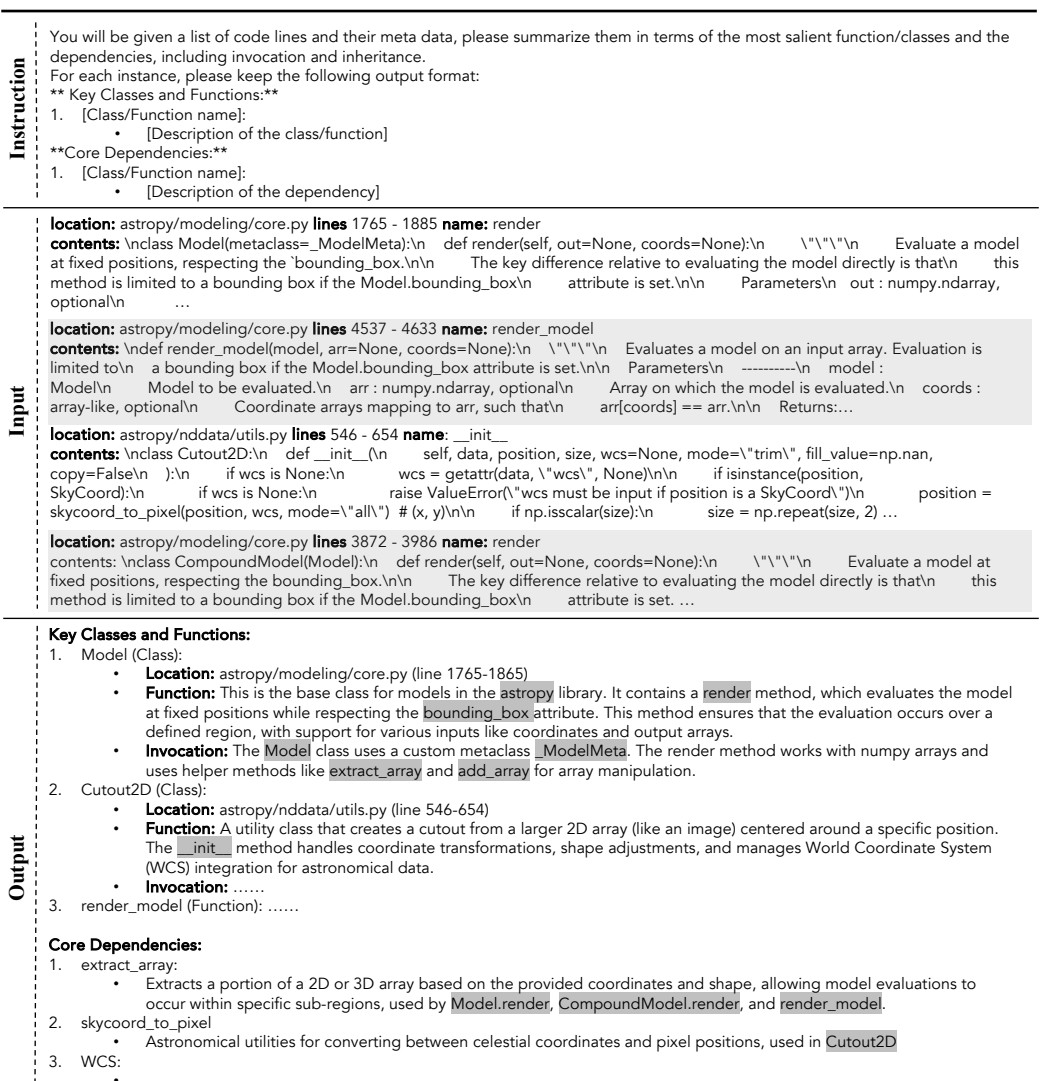

Figure 9: Instruction used for summarizing the flattened ego-graph.

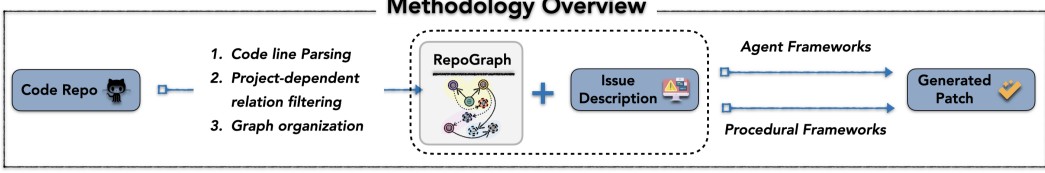

Figure 10: A conceptual overview of our proposed REPOGRAPH.

## C.2    RESULTS ON LOCALIZATION FOR BOTH PASS AND FAIL CASES

In order to highlight the improvement in error localization and correction capability of REPOGRAPH, we provide a thorough analysis of both pass and fail cases. Results are shown in Table 7. Interestingly, we found that the localization accuracy for failure cases is largely lower than the overall average results. This indicates that REPOGRAPH fixes the issues by correctly identifying the editing locations, without which the issue is unlikely to be resolved.

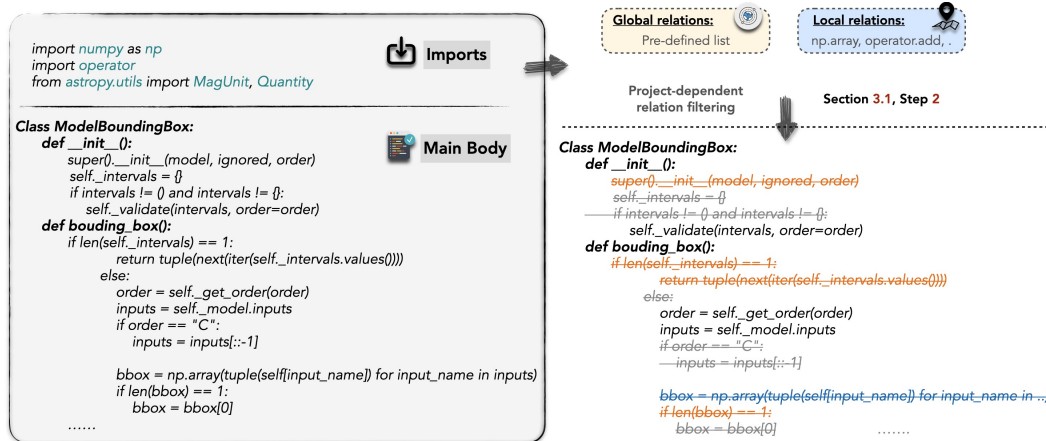

Figure 11: A detailed illustration of how we filter the project-dependent relations given a code snippet. Code line colors are illustrated with respect to global and local relations.

Table 6: Results of REPOGRAPH on SWE-Bench-Lite test set with Claude-3.5-Sonnet in two research lines, including procedural and agent frameworks. We also report the corresponding cost of each method.

| Methods | LLM | Accuracy | | | Avg. Cost | |
|---------|-----|----------|---|---|-----------|---|
| | | *resolve* | *# samples* | *patch apply* | *$ cost* | *# tokens* |
| ***Procedural frameworks*** | | | | | | |
| Agentless | GPT-4o | 27.33 | 82 | 97.33 | $0.34 | 42,376 |
| +REPOGRAPH | GPT-4o | 29.67↑2.34 | 89↑7 | 98.00↑0.67 | $0.39 | 47,323 |
| Agentless | Claude-3.5-Sonnet | 27.67 | 83 | 94.33 | $0.28 | 40,984 |
| +REPOGRAPH | Claude-3.5-Sonnet | 30.33↑2.66 | 91↑8 | 98.67↑4.34 | $ 0.32 | 46,238 |
| ***Agent frameworks*** | | | | | | |
| SWE-agent | GPT-4o | 18.33 | 55 | 87.00 | $2.53 | 498,346 |
| +REPOGRAPH | GPT-4o | 20.33↑2.00 | 61↑6 | 90.33↑3.33 | $2.69 | 518,792 |
| SWE-agent | Claude-3.5-Sonnet | 23.00 | 69 | 86.67 | $1.62 | 521,208 |
| +REPOGRAPH | Claude-3.5-Sonnet | 25.33↑2.33 | 76↑7 | 90.67↑4.00 | $ 1.70 | 539,942 |

## C.3 RESOLVE RATE BY REPOSITORY

We plot the results of the resolve rate in terms of repository distribution in Figure 12. From the figure, it is clear that the resolution of issues varies significantly across different repositories. Notably, the Django and Sympy repositories have the most unresolved issues, with 75 and 61 unresolved issues, respectively. This may indicate a higher level of complexity in the issues or a larger backlog compared to the other repositories. On the other hand, Django has the highest number of resolved issues, with 39 cases. This highlights a strong effort to address issues, even though the unresolved count is still high. Sympy follows closely with 16 resolved issues, suggesting a similar trend. Other repositories like Scikit-learn, Sphinx, and Matplotlib have comparatively fewer issues overall, but their resolve rates are more balanced. For instance, Sphinx shows a ratio of 13 resolved to 3 unresolved issues, reflecting a more consistent effort in issue resolution.

## C.4 RESOLVE RATE BY TIME

We plot the results of the resolve rate in terms of the distribution of releasing time for repositories in Figure 13. Most of the issues were observed in recent years, starting from 2018, with a substantial increase in the total number of issues identified after 2018. In the early years (2012-2016), the number of issues remained relatively low, with both resolved and unresolved counts being minimal. Starting from 2017, there is a noticeable increase in unresolved issues, with 13 unresolved and only 3 resolved issues. In 2019, the number of resolved and unresolved issues significantly increased,

Table 7: Percentage of problems for accurate edition localizations for Agentless+REPOGRAPH with respect to file, function, and line levels. All the numbers are computed from the corresponding generated patches. We also report the accuracy for all pass and failure cases, respectively.

| Methods | LLM | *file* | *function* | *line* |
|---|---|---|---|---|
| Agentless +REPOGRAPH | GPT-4o | 74.3 | 54.0 | 36.7 |
| Pass cases | GPT-4o | 91.0 | 79.8 | 65.2 |
| Failure cases | GPT-4o | 67.3 | 43.1 | 24.6 |

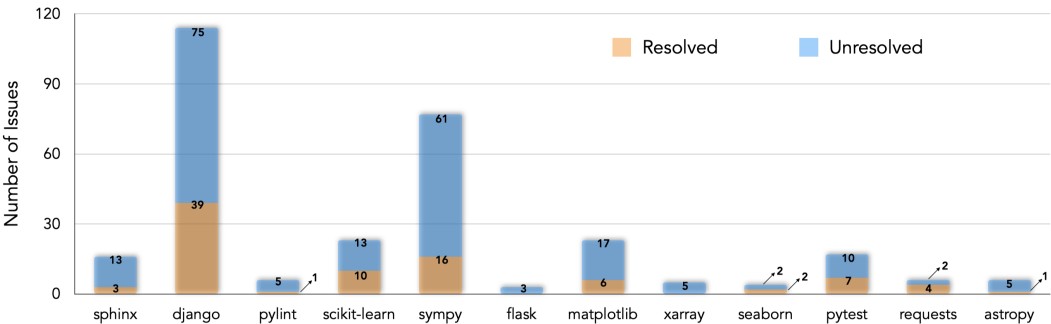

Figure 12: Distribution of issues resolved by Agentless +REPOGRAPH plotted in terms of different repositories.

with 19 resolved out of 59 issues. This trend continued to rise until 2020, where 47 issues remained unresolved, and only 17 were resolved, marking the year with the highest number of unresolved issues in the dataset. By 2021 and 2022, the number of unresolved issues slightly decreased, while the resolve rate increased compared to 2020. This suggests an improvement in the system's ability to address issues in these years. In 2023, although the total number of issues dropped to 30, the proportion of resolved issues remained strong, with 8 resolved out of 22 issues.

# D EXAMPLES FOR ERROR ANALYSES

To help better understand the error category listed in Section 5.4, we provide one example for each category in Figure 14, Figure 15, and Figure 16.

# E LIMITATIONS AND FUTURE WORK

(i) We only explored proprietary LLMs, i.e., GPT-4 series. Due to the poor performance of open-source models on this challenging task, we opted for proprietary models that have demonstrated superior results in code-related tasks. However, a comprehensive evaluation of open-source models such as Llama (Dubey et al., 2024) could be a valuable direction for future work, particularly as these models continue to improve.

(ii) Experiments were only conducted on the Lite set due to the high cost of running large-scale experiments with proprietary models. Exploring more efficient model deployment strategies and alternative cost-effective options for running experiments on larger datasets will be essential for broader applicability.

(iii) Although REPOGRAPH could be adapted to support other programming languages by adjusting the parsing schemes in the implementation, we only explored Python in our main experiments. Future work could extend this approach to other widely used programming languages, such as JavaScript, Java, or C++, to evaluate the generalizability of our methodology across different programming paradigms.

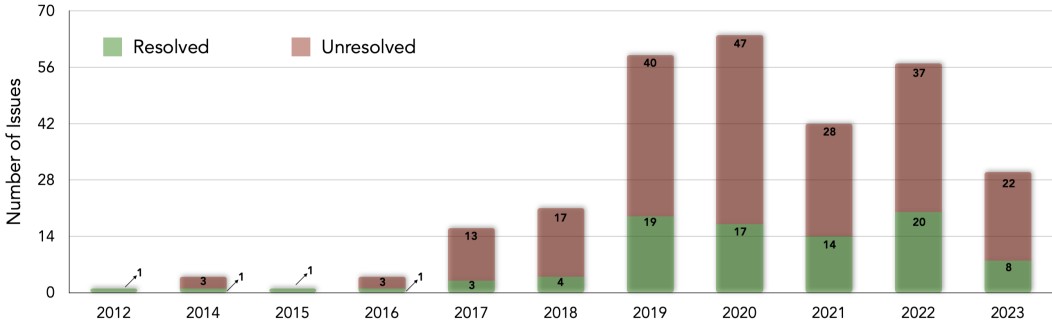

Figure 13: Distribution of issues resolved by Agentless +REPOGRAPH plotted in terms of release time for repositories.

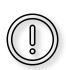

**Issue**

Correct expected format in invalid DurationField error message

**Description**
If you enter a duration "14:00" into a duration field, it translates to "00:14:00" which is 14 minutes. The current error message for invalid DurationField says that this should be the format of durations: "[DD] [HH:[MM:]]ss[.uuuuuu]". But according to the actual behaviour, it should be: "[DD] [[HH:]MM:]ss[.uuuuuu]", because seconds are mandatory, minutes are optional, and hours are optional if minutes are provided. This seems to be a mistake in all Django versions that support the DurationField. Also the duration fields could have a default help_text with the requested format, because the syntax is not self-explanatory.

```
diff --git a/django/forms/fields.py b/django/forms/fields.py
index 935044e..bfb3d28 100644
--- a/django/forms/fields.py
+++ b/django/forms/fields.py
@@ -468,9 +468,10 @@ class DateTimeField(BaseTemporalField):

 class DurationField(Field):
    default_error_messages = {
-      'invalid': _('Enter a valid duration.'),
+      'invalid': _('Enter a valid duration in the format [DD]
[[HH:]MM:]ss[.uuuuuu].'),
      'overflow': _('The number of days must be between
{min_days} and {max_days}.')
      }
+    default_help_text = _('Enter a duration in the format [DD]
[[HH:]MM:]ss[.uuuuuu].')

   def prepare_value(self, value):
      if isinstance(value, datetime.timedelta):
```

**(a) RepoGraph**

```
diff --git a/django/db/models/fields/__init__.py
b/django/db/models/fields/__init__.py
index 77ba99e..de7d841 100644
--- a/django/db/models/fields/__init__.py
+++ b/django/db/models/fields/__init__.py
@@ -1587,7 +1587,7 @@ class DurationField(Field):
      empty_strings_allowed = False
      default_error_messages = {
         'invalid': _("'%(value)s' value has an invalid format.
It must be in "
-                  "[DD] [HH:[MM:]]ss[.uuuuuu] format.")
+                  "[DD] [[HH:]MM:]ss[.uuuuuu] format.")
      }
      description = _("Duration")
```

**(b) Groundtruth**

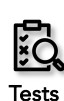

**Tests**

| | |
|---|---|
| test_dumping (model_fields.test_durationfield.TestSerialization) | ✔ |
| test_loading (model_fields.test_durationfield.TestSerialization) | ✔ |
| test_formfield (model_fields.test_durationfield.TestFormField) | ✔ |
| test_exact (model_fields.test_durationfield.TestQuerying) | ✔ |
| test_invalid_string (model_fields.test_durationfield.TestValidation) | ✘ |
| test_gt (model_fields.test_durationfield.TestQuerying) | ✔ |
| test_create_empty (model_fields.test_durationfield.TestSaveLoad) | ✔ |
| test_fractional_seconds (model_fields.test_durationfield.TestSaveLoad) | ✔ |
| test_simple_roundtrip (model_fields.test_durationfield.TestSaveLoad) | ✔ |

Figure 14: An example of *incorrect localization*. The correct patch modifies the "DurationField" in django/db/models/fields/__init__.py. This is the correct place to handle the error message formatting for the "DurationField" used in Django models. REPOGRAPH, however, modifies django/forms/fields.py. This file handles Django form fields, not the model fields. While both model and form fields have overlapping behavior, in this case, the correction is required for the model field (DurationField in __init__.py), not the form field.

## F    IMPACT STATEMENT

The impact of this paper lies in its substantial contribution to enhancing the capability of AI-driven software engineering, particularly with respect to repository-level code understanding. The introduc-

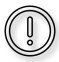

**Issue**

Trailing whitespace in DatasetGroupBy text representation
When displaying a DatasetGroupBy in an interactive Python session, the first line of output contains a trailing whitespace. The first example in the documentation demonstrate this:

```
>>> import xarray as xr, numpy as np
>>> ds = xr.Dataset(
...     {"foo": (("x", "y"), np.random.rand(4, 3))},
...     coords={"x": [10, 20, 30, 40], "letters": ("x", list("abba"))})
>>> ds.groupby("letters")
DatasetGroupBy, grouped over 'letters'
2 groups with labels 'a', 'b'.
```

There is a trailing whitespace in the first line of output which is "DatasetGroupBy, grouped over 'letters' ". This can be seen more clearly by converting the object to a string (note the whitespace before `\n`):

```
>>> str(ds.groupby("letters"))
"DatasetGroupBy, grouped over 'letters' \n2 groups with labels 'a', 'b'."
```

While this isn't a problem in itself, it causes an issue for us because we use flake8 in continuous integration to verify that our code is correctly formatted and we also have doctests that rely on DatasetGroupBy textual representation. Flake8 reports a violation on the trailing whitespaces in our docstrings. If we remove the trailing whitespaces, our doctests fail because the expected output doesn't match the actual output. So we have conflicting constraints coming from our tools which both seem reasonable. Trailing whitespaces are forbidden by flake8 because, among other reasons, they lead to noisy git diffs. Doctest want the expected output to be exactly the same as the actual output and considers a trailing whitespace to be a significant difference. We could configure flake8 to ignore this particular violation for the files in which we have these doctests, but this may cause other trailing whitespaces to creep in our code, which we don't want. Unfortunately it's not possible to just add `# NoQA` comments to get flake8 to ignore the violation only for specific lines because that creates a difference between expected and actual output from doctest point of view. Flake8 doesn't allow to disable checks for blocks of code either.

Is there a reason for having this trailing whitespace in DatasetGroupBy representation? Whould it be OK to remove it? If so please let me know and I can make a pull request.

```
diff --git a/xarray/core/groupby.py b/xarray/core/groupby.py
index bf63803..7f9a25e 100644
--- a/xarray/core/groupby.py
+++ b/xarray/core/groupby.py
@@ -1011,5 +1011,10 @@ class DatasetGroupBy(GroupBy,
ImplementsDatasetReduce):
        return self.map(lambda ds: ds.assign(**kwargs))

+def __repr__(self):
+    # Original representation with trailing whitespace removed
+    groups = "', '".join(str(k) for k in self.groups)
+    return f"DatasetGroupBy, grouped over
{self._group.name}\n{len(self.groups)} groups with labels
'{groups}'."
+ ops.inject_reduce_methods(DatasetGroupBy)
- ops.inject_binary_ops(DatasetGroupBy)
+ ops.inject_binary_ops(DatasetGroupBy)
```

**(a) RepoGraph**

```
diff --git a/xarray/core/groupby.py b/xarray/core/groupby.py
index bf63803..9aac7fc 100644
--- a/xarray/core/groupby.py
+++ b/xarray/core/groupby.py
@@ -436,7 +436,7 @@ class GroupBy(SupportsArithmetic):
        return zip(self._unique_coord.values, self._iter_grouped())

    def __repr__(self):
-       return "{}, grouped over {!r} \n{!r} groups with labels
{}.".format(
+       return "{}, grouped over {!r}\n{!r} groups with labels
{}.".format(
            self.__class__.__name__,
            self._unique_coord.name,
            self._unique_coord.size,
```

**(b) Groundtruth**

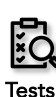

**Tests**

| | |
|---|---|
| xarray/tests/test_groupby.py::test_consolidate_slices | ✓ |
| xarray/tests/test_groupby.py::test_multi_index_groupby_sum | ✓ |
| xarray/tests/test_groupby.py::test_groupby_map_shrink_groups[obj0] | ✓ |
| xarray/tests/test_groupby.py::test_da_groupby_map_func_args | ✓ |
| xarray/tests/test_groupby.py::test_groupby_repr[obj0-x] | ✗ |
| ...... | ... |
| xarray/tests/test_groupby.py::test_groupby_reduce_dimension_error | ✓ |
| xarray/tests/test_groupby.py::test_groupby_repr_datetime[obj0] | ✗ |
| xarray/tests/test_groupby.py::test_groupby_getitem | ✓ |

Figure 15: An example of *contextual misalignment*. The model-generated patch places the `__repr__` method in the wrong class DatasetGroupBy, whereas the correct patch modifies it in the GroupBy class. The `__repr__` method should be implemented in the GroupBy class because it deals with the general group-by functionality. "DatasetGroupBy" is a subclass and doesn't require a new representation method if "GroupBy" already has one. Additionally, when fixing white space trailing, while functionally similar, REPOGRAPH's generation doesn't exactly match the style or intention of the correct patch.

tion of REPOGRAPH not only significantly improves Large Language Models (LLMs) in navigating and comprehending entire codebases, but also showcases the potential of integrating repository-wide structures into AI workflows. By extending the scope from function-level tasks to holistic repository management, REPOGRAPH pushes the boundaries of AI's utility in modern software engineering.

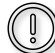

**Issue**

expr.atoms() should return objects with no args instead of subclasses of Atom
`expr.atoms()` with no arguments returns subclasses of `Atom` in `expr`. But the correct definition of a leaf node should be that it has no `.args`.

This should be easy to fix, but one needs to check that this doesn't affect the performance.

```
diff --git a/sympy/core/basic.py b/sympy/core/basic.py
index 97eb81a..4fa2f3c 100644
--- a/sympy/core/basic.py
+++ b/sympy/core/basic.py
@@ -500,15 +500,11 @@ class Basic(metaclass=ManagedProperties):
        {I*pi, 2*sin(y + I*pi)}
-      if types:
-          types = tuple(
-             [t if isinstance(t, type) else type(t) for t in types])
+      if not types:
+          result = {expr for expr in preorder_traversal(self) if not expr.args}
       else:
-          types = (Atom,)
-      result = set()
-      for expr in preorder_traversal(self):
-          if isinstance(expr, types):
-             result.add(expr)
+          types = tuple(t if isinstance(t, type) else type(t) for t in types)
+          result = {expr for expr in preorder_traversal(self) if isinstance(expr,
types)}
       return result
```
**(a) RepoGraph**

```
diff --git a/sympy/core/basic.py b/sympy/core/basic.py
index 97eb81a..f1f411f 100644
--- a/sympy/core/basic.py
+++ b/sympy/core/basic.py
@@ -507,7 +507,7 @@ class
Basic(metaclass=ManagedProperties):
           types = (Atom,)
       result = set()
       for expr in preorder_traversal(self):
-          if isinstance(expr, types):
+          if isinstance(expr, types) and not expr.args:
             result.add(expr)
       return result
```
**(b) Groundtruth**

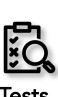

**Tests**

| | |
|---|---|
| test_literal_evalf_is_number_is_zero_is_comparable | ✓ |
| test_subs_with_unicode_symbols | ✓ |
| test_free_symbols_empty | ✓ |
| test_preorder_traversal | ✓ |
| test_atomic | ✗ |
| …… | … |
| test_canonical_variables | ✓ |
| test_sizeof | ✗ |
| test_CommaOperator | ✓ |

Figure 16: An example of *regressive fix*. The model-generated patch successfully resolves the issue which requires the return of objects with no args instead of subclasses of Atom by adding the key code line if not expr.args. However, the fix introduces other new issues such as the size of return objects, as exemplified in the unit tests.

This advancement opens new opportunities for using LLMs in complex engineering tasks such as automated debugging, repository maintenance, and large-scale refactoring. Furthermore, by highlighting the importance of repository-level context for accurate code generation and maintenance, the paper sets a new trajectory for future research in AI and software engineering. It encourages deeper exploration of AI's ability to not only write code but also understand and manage large-scale software projects more efficiently. We foresee minimal risks or negative societal impacts from this work. All datasets and benchmarks used in the evaluation are publicly available, and we adhered to their respective licenses. Additionally, REPOGRAPH has been open-sourced, making it accessible to the research community, particularly to groups with limited access to extensive computing resources, thus fostering broader adoption and further development.

