# OpenReview forum: "RepoGraph: Enhancing AI Software Engineering with Repository-level Code Graph"
_ICLR.cc/2025/Conference — ICLR 2025 Poster_

### Official Review · Reviewer_btxE · 2024-10-21

**Soundness:** 2
**Presentation:** 3
**Contribution:** 3
**Rating:** 5
**Confidence:** 4

**Summary:**

The authors introduce RepoGraph, a new tool that turns a codebase into a graph structure. The authors demonstrate how this tool, which can be integrated into both agent-ic and procedural AI software engineering frameworks, leads to improvements across the board for existing methods. The repository is constructed by filtering out unnecessary file types and repository-dependent code, identifying object oriented programmatic entities, and constructing the remaining pieces of the codebase into a graph. The experimental setup follows existing work. The authors demonstrate improvements for a number of existing AI SWE solutions when RepoGraph is integrated into them and perform some analyses around RepoGraph parameters, localization statistics, and transferability.

**Strengths:**

RepoGraph examines the question of how codebases can be represented effectively. While other works have tried more traditional approaches for localization, such as retrieval and tool-augmented LMs, RepoGraph demonstrates that reshaping the codebase into a graph that is more traversable and useful for a language model can elicit stronger performance on the downstream editing subtask. The authors discuss a pipeline for carrying this out with Python codebases, showing how simple heuristics for removing noise and preprocessing the codebase in a more friendly way can lead to improvements that are method-agnostic.

- [Section 3.1, Step 2] - This step seems quite interesting and novel. I think a visualization, something similar to Figure 2 but a bit more grounded in Step 3’s construction, might be helpful for readers to understand this construction more easily.

**Weaknesses:**

- Figures 1, 2 - I appreciate the effort that went into these figures, they are clearly drawn quite thoroughly. However, I think it is a bit too complex, making it hard to understand. I’d recommend simplifying the diagrams to make the takeaways a bit more obvious. For example, it took me a while to understand the meaning of the colors in Figure 2, (a).
- The design of RepoGraph seems to be written in a fairly Python-centric manner. This is understandable to some degree, as the main evaluation (SWE-bench) is for Python based repositories. However, the contribution would be more impactful if somehow it was demonstrated that this approach scales to other kinds of codebases (with existing work or a new benchmark?). In Section 5.3, the authors mention how for CrossCodeEval, the evaluation is limited to Python. The authors discuss this in the appendix, but I think the generalizability of RepoGraph is a question that should be answered instead of being deferred to future work.
- [Section 3.1, Step 2] - “We maintain a comprehensive list of methods” ⇒ This sounds like it’s hard-coded for Python. Would I have to manually re-perform this filtering for other languages? Is there an automatic filtering criteria I can apply to come up with a comparable list for another language?
- [Section 5.2, Line 361] - This claim seems somewhat handwavy to me. The cost metric seems to go up with every method due to the inclusion of RepoGraph. However, this takeaway seems to suggest that the performance increase is proportional or lower than the increased cost - why is this true? What numbers are being compared for this proportion? (e.g. 2.66% increase vs. 0.04 cost increase? Why is this lower?)
- [Section 5.1, Lines 413 - 420] - The discussion on the differences between procedural and agentic approaches is interesting, but I think it’s not quite clear to me why this discussion is included or how it relates to RepoGraph. Is the suggestion that procedural approaches take more advantage of RepoGraph?

**Questions:**

- [Introduction, Line 75] - “they tend to focus narrowly on specific files, resulting in local optimums”. I’m not sure what this means? Why do agent approaches have this tendency?
- [Intro, Line 85] - What is an “ego-graph”? I see the citation in Line 256, but may be good to define this explicitly as a preliminary somewhere earlier.
- [Section 3.1, Line 205] - how is “non-essential” determined? I could imagine that editing requirements.txt might be necessary to import a new library for carrying out a fix.
- [Section 3.1, Line 216] - What happens in step 1 if the codebase is not written in an object oriented manner? For instance, what if you’re dealing with JavaScript code primarily written in a functional programming style, which may not have classes? Or if you are editing a bash script which may not have these entities?
- [Section 3.2, Line 267] - What is an “edition” stage? Perhaps this is a type? (“Editing”?) Repeated again on Line 350
- [Table 2] - What is # samples?
- [Table 4] - The number of nodes is a bit confusing. In SWE-bench, each codebase has an average of 438k lines. If each node corresponds to a line, why are there only 1419 nodes on average?
- [Figure 3] - Is this for SWE-bench Lite? Or CrossCodeEval?

---

> ### Author Response · Authors · 2024-11-21
> **Response to Reviewer btxE (Part I)**
>
> Thanks for your review and detailed suggestions. We are glad that you liked our design of RepoGraph and think that this method-agnostic plug-in is “traversable, useful, and novel”. Below we provide detailed responses to each of your reviews and hopefully they could help address your concerns.
>
> **W1: Presentation of figures of Section 3.1, Step 2, and Figure 2.**
>
> Thanks for your advice on the figures! For narration in Section 3.1 Step 2, we appreciate your recognition of this design and agree with your comment that an illustration would help readers better understand what’s going on in this step. We have plotted a figure using a specific example in Appendix B.2 due to space limits. For Figure 2, your suggestion makes a lot of sense and we plot a simplified overall pipeline in Figure 10 Appendix B.2 for reference. We also revised the caption of Figure 2 to better help readers understand the meanings of corresponding colors.
>
> **W2: The design of RepoGraph seems to be written in a fairly Python-centric manner. The authors discuss this in the appendix, but I think the generalizability of RepoGraph is a question that should be answered instead of being deferred to future work.**
>
> Thanks for your comment. Theoretically, RepoGraph could be applied to other programming languages. Specifically for construction, Step 1, “code line parsing,” could be smoothly extended to other programming languages by using different versions of AST. For example, one could use “Acorn” for Javascript. Step 2, “project-dependent relation filtering,” could also be realized in other languages with details mentioned in W3. After RepoGraph construction, LLMs should naturally leverage the information in the structure since they have been exposed to code in various programming languages. We only conduct Python-related experiments in this paper since this is the most popularly adopted language in previous works, with the modification discussed earlier, we believe that RepoGraph can be applied to other programming languages as well.
>
> **W3: [Section 3.1, Step 2] - “We maintain a comprehensive list of methods” ⇒ This sounds like it’s hard-coded for Python. Would I have to manually re-perform this filtering for other languages? Is there an automatic filtering criteria I can apply to come up with a comparable list for another language?**
>
> Thanks for your question. We agree that we will need some modifications when shifting to other programming languages. However, the list is not fully hard-coded. For example, in Python, we use methods such as ``sys.builtin_module_names`` and ``pkgutil`` to find the built-in and standard libraries. Apart from these, we also include the default methods for certain data structures such as ``list`` and ``tuple``. Similar approaches or methods of other programming languages could be leveraged to form a semi-automatic curation approach for such a list.
>
> **W4: [Section 5.2, Line 361] - This claim seems somewhat handwavy to me. The cost metric seems to go up with every method due to the inclusion of RepoGraph. However, this takeaway seems to suggest that the performance increase is proportional or lower than the increased cost - why is this true? What numbers are being compared for this proportion? (e.g. 2.66% increase vs. 0.04 cost increase? Why is this lower?)**
>
> Thanks for your observation. We agree that the current expression might be a little ambiguous to readers. Our intention is to highlight that the additional token cost brought by RepoGraph is acceptable with such performance improvement, especially compared with the huge cost with previous methods such as SWE-agent. We have modified this sentence in the revised PDF.
>
> **W5: [Section 5.1, Lines 413 - 420] - The discussion on the differences between procedural and agentic approaches is interesting, but I think it’s not quite clear to me why this discussion is included or how it relates to RepoGraph. Is the suggestion that procedural approaches take more advantage of RepoGraph?**
>
> By providing such analyses, we aim to show how RepoGraph performs when integrating with agent and procedural frameworks, especially on “localization.” The discussion here is to provide insights for readers and potential users of RepoGraph so that they know how to fully take advantage of RepoGraph when integrating with their own frameworks.

---

> ### Author Response · Authors · 2024-11-21
> **Response to Reviewer btxE (Part II)**
>
> **Q1: [Introduction, Line 75] - “they tend to focus narrowly on specific files, resulting in local optimums”. I’m not sure what this means? Why do agent approaches have this tendency?**
>
> Thanks for your question. Compared with procedural frameworks or our RepoGraph, agent frameworks rely on specific action designs to navigate the whole repository, without a holistic view of repository structure. For example, agent frameworks will easily locate a certain Python file with the command “open xx.py” and go to specific code lines using “goto #num.” However, they cannot grasp the correlations or code lines/files. Therefore, agent frameworks might go around in circles. That’s why intuitively RepoGraph could help agent frameworks by providing an overview of repository structures as a navigation.
>
> **Q2: [Intro, Line 85] - What is an “ego-graph”? I see the citation in Line 256, but may be good to define this explicitly as a preliminary somewhere earlier.**
>
> Thanks for your comments. We agree that an explicit explanation of “ego-graph” will greatly help readers understand. We have added these in the introduction section. Please check the updated PDF.
>
> **Q3: [Section 3.1, Line 205] - how is “non-essential” determined? I could imagine that editing requirements.txt might be necessary to import a new library for carrying out a fix.**
>
> Thanks for your question. Our point here is that we want to focus on the relevant code file and exclude those files without code. We agree that “non-essential” might be a little misleading, and we have already changed the expression. We also appreciate your comment on the example of “requirements.txt,” which could be useful for fixes. While our paper currently focuses on code files only, we will actively explore this insight by leveraging other file types in the future.
>
> **Q4: [Section 3.1, Line 216] - What happens in step 1 if the codebase is not written in an object oriented manner? For instance, what if you’re dealing with JavaScript code primarily written in a functional programming style, which may not have classes? Or if you are editing a bash script which may not have these entities?**
>
> Thanks for your insightful question. For functional programming styles such as JavaScript, RepoGraph could be naturally extended to this setting since “classes” and “functions” are both building units of RepoGraph. Even if the code does not have classes, RepoGraph is able to represent the relations of code lines within functions and across functions. The same applies to the case of bash script if the script contains dependencies.
>
> **Q5: [Section 3.2, Line 267] - What is an “edition” stage? Perhaps this is a type? (“Editing”?) Repeated again on Line 350**
>
> Thanks for the catch! It actually means “code editing”. We have fixed the typos in both places in the newly updated PDF.
>
> **Q6: [Table 2] - What is # samples?**
>
> \# samples is the number of correctly resolved data points in SWE-Bench-Lite. For example, 89 means a total of 89 data points resolved out of 300 test cases.
>
> **Q7: [Table 4] - The number of nodes is a bit confusing. In SWE-bench, each codebase has an average of 438k lines. If each node corresponds to a line, why are there only 1419 nodes on average?**
>
> Thanks for your question. As you have observed in Section 3.1 Step 2, we only considered code lines (nodes) that contain project-related calling relations. Although the original number of code lines in SWE-bench could be very large, many of the code lines are actually comments or assignment statements that do not contain function calls. Among code lines with function calls, most of them leverage built-in or third-party methods, which is not our major focus. Therefore, with the filtering process used in RepoGraph, we will filter most of the code lines out and leave only the core context to construct the graph structure.
> Of course, content filtered out, such as code comments, could be of potential use for repo-level tasks. We leave this exploration as future work, and they could also be a seamless add-on to RepoGraph as a heterogeneous structure.
>
> **Q8: [Figure 3] - Is this for SWE-bench Lite? Or CrossCodeEval?**
>
> The results are for SWE-bench-Lite. We have updated this info in the newly updated PDF.
>
> We sincerely appreciate your comments which helped improve our paper and we hope our responses above helped to address your concerns. Please feel free to leave us more comments if you have and we are always happy to engage in discussions.

---

> ### Author Response · Authors · 2024-12-01
>
> Dear Reviewer btxE,
>
> Thank you for dedicating your time and effort to reviewing our work. We understand that your schedule may be quite busy and we truly appreciate your thoughtful evaluation. As the author-reviewer discussion phase approaches its end, we kindly request your feedback on whether our response has effectively addressed your concerns and if there are any additional questions or points you would like to discuss.
>
> Looking forward to further discussion and your valuable insights.
>
> Best regards,
>
> Paper 8051 Authors

---

> > ### Comment · Reviewer_btxE · 2024-12-02
> > **Response to Authors**
> >
> > Thanks for your thorough responses, I greatly appreciate the time and effort that went into responding to each of the points.
> >
> > I have decided to maintain my score. While I really appreciate the answers, I feel like the extensibility of RepoGraph to different programming languages is still something that should be addressed more directly. I agree with the authors that conceptually, this framework might be easily extensible to non-Python methods. However, based on my reading of the code and understanding of the work, I maintain my position that the way RepoGraph is designed still feels quite tethered to Python. Most significantly, replacing Python-oriented heuristics of adapting the codebase to a format that can be converted into the graph structure discussed in the paper feels non-trivial. I think the primary analyses I would want to see is how RepoGraph + (system) performs on a non-Python codebase-level evaluation, although I do understand that not many of these evaluations exist (Defects4J, SWE-bench Multimodal)

---

> ### Author Response · Authors · 2024-12-03
> **Thanks for your reply**
>
> Thank you for your response and for reviewing our code. Due to the last-minute rebuttal period, we are unable to conduct a systematic evaluation of other programming languages at this time. However, we would like to provide detailed instructions on how to adapt RepoGraph to other programming languages with minimal code changes.
>
> The core components, particularly the parsers, can be easily adjusted. Here are the steps:
>
> 1. Update the language configuration for the parsers: In repograph/construct_graph.py (line 238), download the grammar for the target language and place it in the appropriate directory. For instance, for Java, the grammar can be obtained from [tree-sitter-java](https://github.com/tree-sitter/tree-sitter-java/blob/master/queries/tags.scm).
> 2. Update the AST parser for the target programming language: Configure the parser for the desired language in construct_graph.py (line 283). For instance, you could use the `javalang` library for Java.
>
> These changes can be consolidated into a configuration file, allowing them to be implemented once and reused as needed. It only takes a non-expert a few hours to adapt RepoGraph to corresponding programming languages, and we expect it to take much less time for the developers.
>
> We would also like to kindly point out that several Python-centric approaches [1,2,3] have been accepted into prestigious conferences like ICLR. Therefore, we respectfully hope that this aspect does not weigh heavily on the overall evaluation of our work.
>
> [1] Jain, Naman, et al. "Llm-assisted code cleaning for training accurate code generators." ICLR 2024.
>
> [2] Zhang, Shun, et al. "Planning with large language models for code generation." ICLR 2023.
>
> [3] Carlos E. Jimenez, et al. “SWE-bench Can Language Models Resolve Real-World GitHub Issues?” ICLR 2024.

---

### Official Review · Reviewer_wovV · 2024-10-25

**Soundness:** 4
**Presentation:** 4
**Contribution:** 4
**Rating:** 8
**Confidence:** 5

**Summary:**

This paper introduces a framework called *RepoGraph* for fine-grained modeling of code repositories to address the current challenges in repository-level code tasks. *RepoGraph* leverages tree-sitter to model code repositories at the level of code behavior, constructing a code structure graph based on the invocation and dependency relationships between functions and classes. By using this graph, *RepoGraph* provides more accurate code context for given retrieval terms, thereby improving the accuracy of code task completion. The authors tested *RepoGraph* on the **SWE-bench** benchmark, demonstrating that it can be seamlessly integrated with both advanced agent-based and non-agent frameworks, significantly enhancing the accuracy of issue-fixing tasks. Furthermore, they tested it on the **CrossCodeEval** benchmark, verifying that *RepoGraph* can also be applied to repository-level code completion tasks.

**Strengths:**

1.	Conducting high-quality, fine-grained modeling of code repositories is extremely challenging due to the numerous corner cases that need to be considered. Implementing such a detailed model for code repositories is no easy task, and I truly appreciate the authors’ efforts.
2.	*RepoGraph* can be seamlessly integrated with both agent-based and non-agent frameworks, significantly improving the ability to fix issues on the **SWE-bench**. This will be highly beneficial for future agent-driven code development.
3.	*RepoGraph* can also be applied to repository-level code completion tasks, which will be very useful for code completion in daily development.
4.	The authors conducted extensive experiments to validate the effectiveness of *RepoGraph*.
5.	Compared to other papers I have reviewed, this paper is well-written, with a clear structure and a balanced level of detail.

**Weaknesses:**

This paper is well-written and does not have any major weaknesses, but there are a few minor typos:

•	`Line 228`: There is a repetition of `such as`.

•	`Line 306`: The sentence `To assess the patch application rate, we attempt to apply the generated patches to the repository using the patch program; successful applications contribute to this metric.` has a somewhat disjointed structure and logic. The semicolon is typically used to connect two independent but related clauses, but in this case, the first part of the sentence is not a complete clause, even though there is a logical connection between the two parts.

**Questions:**

1.	Although the authors mentioned that due to resource limitations, only 500 samples were used for experiments on **CrossCodeEval**, the cost of conducting a full test on **CrossCodeEval** does not seem to be significantly higher compared to testing on **SWE-Bench**. This is because the cross-file context retrieved should not be very long. Could the authors report the costs of each test?
2.	While GPT-4o is also capable of code completion, **CrossCodeEval** primarily evaluates the completion abilities of base code models. Could the authors include results from some open-source code models, such as `Deepseek-Coder-V2` and `Qwen2.5-Coder`? These models should be relatively easy to run locally for inference.

---

> ### Author Response · Authors · 2024-11-21
> **Response to Reviewer wovV**
>
> Thanks for your strong review and the insightful comments. We feel encouraged and genuinely appreciate your recognition of our  "extremely challenging" problem setting, "highly beneficial" method, "extensive" experiments, and "well-written" paper.
>
> **W1: minor typos in line 228 and line 306.**
>
> Thanks for the catch. We have fixed the two typos correspondingly. Please refer to line 227 and line 306 in the newly updated revised PDF.
>
> **Q1: Although the authors mentioned that due to resource limitations, only 500 samples were used for experiments on CrossCodeEval, the cost of conducting a full test on CrossCodeEval does not seem to be significantly higher compared to testing on SWE-Bench. This is because the cross-file context retrieved should not be very long. Could the authors report the costs of each test?**
>
> Thanks for your comment. We initially only used 500 samples for experiments due to the time and fund limit before submission. After the submission deadline, we continued to finish the experiments on all 2,665 samples, and the final results are updated in the following table. We also report the average cost for CroseCodeEval using GPT-4o for references.
>
> |             | Codematch-EM | Codematch-ES | Identifier-EM | Identifier-ES | Avg. Cost |
> |-------------|--------------|--------------|---------------|---------------|-----------|
> | GPT-4o      | 10.5         | 59.6         | 16.8          | 47.9          | 1.53k     |
> | +RepoGraph  | 28.7         | 68.9         | 36.0          | 61.3          | 9.24k     |
>
> **Q2: While GPT-4o is also capable of code completion, CrossCodeEval primarily evaluates the completion abilities of base code models. Could the authors include results from some open-source code models, such as Deepseek-Coder-V2 and Qwen2.5-Coder? These models should be relatively easy to run locally for inference.**
>
> Thanks for the insightful question about open-source code models. We have conducted experiments on CrossCodeEval with Deepseek-Coder-V2-Lite-Instruct (locally deployed). The results are shown in the following:
>
> |                     | Codematch-EM | Codematch-ES | Identifier-EM | Identifier-ES |
> |---------------------|--------------|--------------|---------------|---------------|
> | Deepseek-Coder-V2   | 10.2         | 57.3         | 16.6          | 49.1          |
> | +RepoGraph          | 19.7         | 67.8         | 29.7          | 59.2          |
>
>
> We found that open-source models, trained specifically for code intelligence, such as Deekseek-Coder, demonstrate very similar performance with powerful proprietary models like GPT-4o. When integrating with RepoGraph, we also observe a consistent performance gain. However, we find that the improvement brought by RepoGraph is more pronounced when integrated with GPT-4o. A potential reason is that GPT-4o is better at interpreting the additional context information provided for better code reasoning.
> We have updated the new results and findings in Section 5.3 of the newly updated PDF.
>
> We sincerely thank you again for your efforts in reviewing which helped a lot in improving our paper. We hope our responses above could address your concern and please feel free to leave any comments if you have. We are always happy and open to discussions.

---

> > ### Comment · Reviewer_wovV · 2024-11-21
> > **Response to Authors [Good Work]**
> >
> > Thank you for your response. I believe the experiments in this work are thorough, and their research will contribute to the development of advanced code intelligence systems. Therefore, I maintain my score to support the acceptance of this work.

---

### Official Review · Reviewer_v4kH · 2024-11-01

**Soundness:** 2
**Presentation:** 3
**Contribution:** 2
**Rating:** 6
**Confidence:** 4

**Summary:**

This paper introduces RepoGraph, a plug-in designed to enhance the capabilities of LLMs in understanding and navigating code repositories. RepoGraph constructs a graph-based representation of the entire codebase using abstract syntax tree parsing. The system is evaluated using the SWE-bench Lite and CrossCodeEval benchmarks, demonstrating its effectiveness in issue resolution and fault localization. The authors also conduct various analyses, including complementarity and failure cause analysis.

**Strengths:**

1. Originality: RepoGraph introduces a novel approach by leveraging repository-level graph-based representations, which provides a more comprehensive understanding of the codebase.

2. Quality: The paper is well-written, with a clear explanation of how RepoGraph work. It provides a robust experimental setup that evaluates the tool's performance across multiple dimensions.

3. Clarity: The distinctions between RepoGraph and other repository understanding techniques are clearly articulated, making it easy to understand the motivation and design of the tool.

4. Significance: Its contributions could lead to improvements in the effectiveness of automated issue resolution and code navigation tools.

**Weaknesses:**

1. Performance Gains: While RepoGraph shows some improvement in issue resolution and fault localization, the gains are relatively small.

2. Cost Assessment: The paper does not adequately address the potential computational costs associated with graph construction and search operations. A thorough analysis of these costs, including time overhead, would strengthen the paper by providing a better understanding of RepoGraph’s practical feasibility.

3. Reproducibility: The open-source code provided in the repository is not entirely reproducible based on the available documentation. Specific issues include:

	(1) Environment Setup: The steps for setting up the execution environment are incomplete. It is unclear whether specific configurations or environments are needed for the tool to function properly.

	(2) Configuration: The LLMs require additional keys for configuration, but the paper lacks guidance on how to set these up.

	(3) Evaluation Steps: There is no clear documentation on how to evaluate the generated results effectively. Detailed instructions on the 	evaluation process would help ensure that users can reproduce the experimental findings.

**Questions:**

- Can the authors provide more information about the computational overhead involved in constructing and using the repository-level graphs? Specifically, how does the time complexity compare to other methods?

- Given the small performance improvements, could the authors discuss potential areas for further optimization or enhancement of RepoGraph that could lead to more significant gains?

- Could the authors offer guidance on the reproducibility, specifically:

    - What are the exact environment requirements for running the tool?
    - How should the necessary keys for LLM configuration be set up?
    - What steps should users follow to accurately evaluate the results generated by RepoGraph?

---

> ### Author Response · Authors · 2024-11-21
> **Response to Reviewer v4kH**
>
> Thanks for your detailed review and suggestions. We appreciate your recognition of RepoGraph as “a novel approach”, with a “well-written and clear” presentation and “clearly-articulated” motivations. We provide responses to each of your reviews, and hopefully they could help address your concerns.
>
> **W1 & Q2: Given the small performance improvements, could the authors discuss potential areas for further optimization or enhancement of RepoGraph that could lead to more significant gains?**
>
> Thanks for your comment. While the performance improvement of RepoGraph might seem small, the gain is already comparable with previous frameworks [1, 2]. We also want to highlight that RepoGraph actually significantly improves the localization performance compared with baselines, as shown in Table 3. The final performance also largely depends on the capability of foundational LLMs, which is not our primary focus. As a plug-in method, RepoGraph is universally applicable and only requires a little additional token/time cost, which leads to a pronounced improvement.
>
> As you have mentioned, RepoGraph is an initial attempt in this research line, and there is a lot more to improve performance further. For example, (i) we could leverage other file types in the repository, such as code docs (README), and configurations (requirements.txt); (ii) inside a code file, the code comments and other declarative statements could be further leveraged; and (iii) design dynamic sub-graph retrieval and trimming methods to better integrate RepoGraph with existing methods. We believe that our work will provide a valuable foundation for future research along these directions.
>
> [1] Liu, Xiangyan, et al. "Codexgraph: Bridging large language models and code repositories via code graph databases." arXiv preprint arXiv:2408.03910 (2024).
>
> [2] Zhang, Yuntong, et al. "Autocoderover: Autonomous program improvement." Proceedings of the 33rd ACM SIGSOFT International Symposium on Software Testing and Analysis. 2024.
>
>
> **W2 & Q1: Cost Assessment: The paper does not adequately address the potential computational costs associated with graph construction and search operations. A thorough analysis of these costs, including time overhead, would strengthen the paper by providing a better understanding of RepoGraph’s practical feasibility.**
>
> Thanks for your insightful question. We report the average time for graph construction and search operations for all 300 repositories here in the following table:
>
> |                     | Avg. time |
> |---------------------|-----------|
> | Construction Step 1 | 0.018s    |
> | Construction Step 2 | 7.291s    |
> | Construction Step 3 | 2.364s    |
> | Search              | 1.317s    |
>
>
> We can see that most of the time for construction is for Step 2, i.e., filter project-dependent relations. However, compared with other methods such as SWE-agent, which usually requires several minutes to resolve a specific issue, the time required to construct RepoGraph is negligible.
>
> **W3 & Q3: Reproducibility: The open-source code provided in the repository is not entirely reproducible based on the available documentation.**
>
> Thanks for your constructive comments. We answer your questions regarding three issues respectively in the following:
>
> (i) Environment setups: We already have a requirements.txt listing the detailed versions of all used packages for reproducibility in the anonymous code repository. Apart from the original requirements, for evaluation, we also need to set python=3.11 and swebench=1.1.0. Therefore, we added corresponding install instructions for the environment setup based on that as follows:
> > pip install -r requirements.txt
>
> (ii) Configuration: For integrating RepoGraph with SWE-agent, we follow the instructions in its initial implementation, and the keys to LLMs could be configured in `SWE-agent/keys.cfg`. For Agentless, the keys are directly configured by using the command `export OPENAI_API_KEY=xxxx`.
>
> (iii) Evaluation setups: The evaluation uses the open-source SWE-bench-docker repository (https://github.com/aorwall/SWE-bench-docker). To start evaluation, we first clone the repository to local. Then we need to prepare the outputs generated by the system (one generated patch per instance) and run the corresponding scripts in SWE-bench-docker, which will directly produce the evaluation results.
> > mkdir logs
>
> > chmod 777 logs
>
> > python run_evaluation.py --predictions_path "./all_preds.jsonl"  --log_dir "./logs"  --swe_bench_tasks "princeton-nlp/SWE-bench_Lite"
>
> We will modify our open-sourced code repository correspondingly following your suggestions.
>
> Thanks again for your review to make this paper better. We hope our responses above could address your concerns and please free feel to leave more questions if you have. We are always happy to engage in discussions.

---

> > ### Comment · Reviewer_v4kH · 2024-11-24
> >
> > Thank you for your response. I improved my score.

---

### Official Review · Reviewer_H4KK · 2024-11-04

**Soundness:** 3
**Presentation:** 4
**Contribution:** 4
**Rating:** 6
**Confidence:** 4

**Summary:**

This paper presents an approach to navigating software repos and
offer the desired navigation by incorporating
information at the line level, file level, and repository level. Toward that end, the authors
proposed a graph-learning-based framework with nodes corresponding to code-line level and edges corresponding
to connections of the nodes at the repo level.

**Strengths:**

- A plug-in-based framework to easily integrate with other AI-based tools
- Code and docker provided for reproducibility

**Weaknesses:**

- No human evolution studies
- Missing prior relevant literature

The authors could verify their assessment on SWE-bench Verified, released as a part of the benchmark.

Although the proposed work seems promising, the article lacks relevant literature citations. How to navigate the software repository has been studied by software engineering researchers before the LLM era. The
approaches include the navigation behavior of developers using eye-tracking data, during code summarization tasks.
I encourage the authors to consult the works of Bonita Sharif and follow the references.

References
1. Begel, A., Khoo, Y.P. and Zimmermann, T., 2010, May. Codebook: discovering and exploiting relationships in software repositories. In Proceedings of the 32nd ACM/IEEE International Conference on Software Engineering-Volume 1 (pp. 125-134).
2. Bairi, R., Sonwane, A., Kanade, A., Iyer, A., Parthasarathy, S., Rajamani, S., Ashok, B. and Shet, S., 2024. Codeplan: Repository-level coding using llms and planning. Proceedings of the ACM on Software Engineering, 1(FSE), pp.675-698.

**Questions:**

According to Figure 3, RepGraph was not able to find all error cases when added as a plug-in with Agentless and SWE-Agent.
As RepoGrpah works at line-level, file-level, and repo-level and was added as a plug-in, I expected RepoGrapgh to cover all error cases of Agentless and SWE-Agent.
Do the authors have any intuition or hypothesis of the observation of Figure 3?

---

> ### Author Response · Authors · 2024-11-21
> **Response to Reviewer H4KK**
>
> Thanks for your positive review and detailed suggestions. We appreciate your recognition of RepoGraph’s ease of integration with existing methods and reproducibility. We provide a point-by-point response and hopefully address your concerns.
>
> **W1: No human evaluation studies. The authors could verify their assessment on SWE-bench Verified, released as a part of the benchmark.**
>
> Thanks for your constructive comment. We are contacting the authors of SWE-bench to verify our results and will update it in the paper as soon as it finishes.
>
> **W2: Missing prior relevant literature. Although the proposed work seems promising, the article lacks relevant literature citations. How to navigate the software repository has been studied by software engineering researchers before the LLM era. The approaches include the navigation behavior of developers using eye-tracking data, during code summarization tasks. I encourage the authors to consult the works of Bonita Sharif and follow the references.**
>
> Thanks for pointing these related works out. We have carefully studied these two works and incorporated the related works in the software engineering community to make our literature coverage comprehensive. Please refer to our updated “related work” section in the revised PDF.
>
> **Q1: According to Figure 3, RepGraph was not able to find all error cases when added as a plug-in with Agentless and SWE-Agent. As RepoGrpah works at line-level, file-level, and repo-level and was added as a plug-in, I expected RepoGrapgh to cover all error cases of Agentless and SWE-Agent. Do the authors have any intuition or hypothesis of the observation of Figure 3?**
>
> Thanks for your question. The core mission of RepoGraph is actually to provide a more comprehensive context of the repository to LLMs so that the LLMs have a better chance to pinpoint the exact localization of the current issue. This functionality actually is proved as RepoGraph largely reduces the error case of “incorrect localization” in Figure 3. However, the editing process currently fully relies on the capability of LLMs themselves. That being said, we have no other control over the actual editing process. That’s why RepoGraph may still have errors like “regressive fix” or “contextual misalignment.” We have discussed this in lines 521-524. We present Figure 3 in our work, hoping to offer some insights for future works to further boost the performance.
>
> Thanks again for your review in helping improve our paper and please feel free to leave us more comments if you have, we are happy to engage in discussions.

---

> > ### Comment · Reviewer_H4KK · 2024-11-26
> >
> > Thanks to the authors for their explanation. I do not have further questions.

---

### Official Review · Reviewer_cQhV · 2024-11-04

**Soundness:** 3
**Presentation:** 3
**Contribution:** 3
**Rating:** 6
**Confidence:** 3

**Summary:**

This paper proposes RepoGraph, a plug-in module designed to extract repository-level definition and invocation information, offering graphical navigation to improve AI software engineering. RepoGraph is compatible with popular agent-based and procedural based LLM frameworks. The authors conduct experiments with GPT-4 and GPT-4o on SWE-bench, yielding a promising result.

**Strengths:**

- RepoGraph can be easily built and integrated into existing methods and its construction process is straightforward and well-defined.

- The analysis presented in this paper is clear and detailed. The authors offering several key observations and insightful explanations, which could be inspiring for future research and broader adoption.

- The author conducted additional transferibility experiments to better explain the effectiveness of RepoGraph.

- The authors give comparative analysis between procedural and agent-based methods.

- A thorough classification analysis of error cases is provided, with specific examples for each category.

**Weaknesses:**

- The primary experimental results are presented solely on SWE-bench-Lite, and the improvement in accuracy resolution are not particularly significant. Additionally, the paper mainly focuses on the resolved numbers and accuracy as evaluation criterion. In cases with a limited number of passes, the quantitative comparison and analysis is not convincing enough and may exhibit considerable variance. It would be beneficial for the authors to provide a thorough analysis of both pass and fail cases, as examining failed cases could also highlight improvements in error localization and correction capability of RepoGraph.

- As the authors note, the paper only evaluates based on GPT-4 and GPT-4o. Given the increasing availability of various General LLMs and Code LLMs, such as Claude and other open-source models, it would be better for the authors to include results from these additional LLMs. Previous studies indicate that different LLMs have varying performance when applied to new methods or tasks. It would be valuable to see how RepoGraph performs across a broader range of LLMs.

- A presentation issue in Figure 10: some numbers overlap, and certain white numbers blend into the background.

**Questions:**

The authors state that in the error analysis 'when integrated with REPOGRAPH, the proportion of error types “incorrect localization” and “contextual misalignment” largely decreases'. However, this conclusion appears limited in significance due to the small number of passed samples involved. Could the authors provide further clarification on how this conclusion was reached? Have the authors considered the implications of having too few passed samples in their analysis?

---

> ### Author Response · Authors · 2024-11-21
> **Response to Reviewer cQhV (Part I)**
>
> Thank you for the positive feedback and encouraging comments. We sincerely appreciate your recognition of RepoGraph's straight and well-defined design, ease of integration, as well as insightful experiments and analyses. Below, we provide detailed responses to each of your comments and hope to address any further considerations you may have:
>
> **W1: The primary experimental results are presented solely on SWE-bench-Lite, and the improvement in accuracy resolution is not particularly significant. Additionally, the paper mainly focuses on the resolved numbers and accuracy as evaluation criterion. In cases with a limited number of passes, the quantitative comparison and analysis is not convincing enough and may exhibit considerable variance. It would be beneficial for the authors to provide a thorough analysis of both pass and fail cases, as examining failed cases could also highlight improvements in error localization and correction capability of RepoGraph.**
>
> Thanks for your suggestion. Following your advice, we conducted another round of analysis on the error cases of RepoGraph integrated with Agentless. Specifically, we calculate the accuracy of localization for all error cases:
>
> |                          | file-level | function-level | line-level |
> |--------------------------|------------|----------------|------------|
> | Agentless+RepoGraph      | 74.3       | 54.0           | 36.7       |
> | Pass cases               | 91.0       | 79.8           | 65.2       |
> | Fail cases               | 67.3       | 43.1           | 24.6       |
>
> Interestingly, we found that the localization accuracy for failure cases is significantly lower than average in Table 3. This indicates that RepoGraph fixes the issues by correctly identifying the editing locations, without which the issue is unlikely to be resolved. Even though there are many fail cases, adding RepoGraph could better localize file/function/line, but due to LLM’s correction ability, its final output patch does not pass the test. We have added the results in Appendix C.2.
>
> **W2: As the authors note, the paper only evaluates based on GPT-4 and GPT-4o. Given the increasing availability of various General LLMs and Code LLMs, such as Claude and other open-source models, it would be better for the authors to include results from these additional LLMs. Previous studies indicate that different LLMs have varying performance when applied to new methods or tasks. It would be valuable to see how RepoGraph performs across a broader range of LLMs.**
>
> Thanks for this constructive comment. Following your advice, we additionally use Claude-3.5-Sonnet (claude-3-5-sonnet-20240620) for experiments on SWE-Bench-Lite, and present the results in the following table:
>
> |               | Pass rate | Patch application rate |
> |---------------|-----------|-------------------------|
> | SWE-agent     | 23.00     | 86.67                  |
> | +RepoGraph    | 25.33     | 90.67                  |
> | Agentless     | 27.67     | 94.33                  |
> | +RepoGraph    | 30.33     | 98.67                  |
>
> We also conducted experiments with smaller open-source code LLMs on CrossCodeEval, e.g., Deepseek-Coder-V2-Lite-Instruct (16B), deployed locally. The results are as follows:
>
> |                     | Codematch-EM | Codematch-ES | Identifier-EM | Identifier-ES |
> |---------------------|--------------|--------------|---------------|---------------|
> | Deepseek-Coder-V2   | 10.2         | 57.3         | 16.6          | 49.1          |
> | +RepoGraph          | 19.7         | 67.8         | 29.7          | 59.2          |
>
> We found that RepoGraph is also compatible with other proprietary and open-source code LLMs. We have updated these new results and the corresponding analyses in Appendix C.1 and Section 5.3 correspondingly in the updated PDF version.

---

> ### Author Response · Authors · 2024-11-21
> **Response to Reviewer cQhV (Part II)**
>
> **W3: A presentation issue in Figure 10: some numbers overlap, and certain white numbers blend into the background.**
>
> Thanks for pointing this out and for reading our appendix! We have already fixed this issue to bring out a clearer version for readers. Please check the newer PDF version for the revision of Figures 10 and 11.
>
>
> **Q1: The authors state that in the error analysis 'when integrated with REPOGRAPH, the proportion of error types “incorrect localization” and “contextual misalignment” largely decreases'. However, this conclusion appears limited in significance due to the small number of passed samples involved. Could the authors provide further clarification on how this conclusion was reached? Have the authors considered the implications of having too few passed samples in their analysis?**
>
> Thanks for your question. The current version we present in Figure 3, from our perspective, carries the most insightful takeaways for users and for future researchers. We agree that the number of data points for this analysis could be limited. However, we do observe an overall decreasing trend for all three error types when integrating with RepoGraph. Therefore, we revised the original conclusion to make it more rigorous, as shown in lines 523-524 of the current PDF version.
>
> We sincerely appreciate your review in helping improve our paper and hope our response could be satisfactory. Please feel free to leave us more comments and we are happy to engage in discussions.

---

> > ### Comment · Reviewer_cQhV · 2024-11-25
> >
> > Thanks for the detailed explanation and the added experiments. I think most of my concerns are addressed. I appreciate the authors' effort in sharing their thorough analysis and valuable insights. Thanks!

---

### Author Response · Authors · 2024-11-22
**Thank you for the review and a kind reminder.**

Dear reviewers:

We genuinely appreciate your constructive review and feedback, which helped us to improve our work. We also thank reviewer _wovV_ for the prompt response and firm support of our work. We want to start by expressing our appreciation for the positive recognition of the strengths of our study, including:

- Our proposed RepoGraph is novel (`v4kH`, `btxE`), well-motivated (`v4kH`), and beneficial/easy to integrate with existing methods (`cQhV`, `H4KK`, `wovV`, `btxE`).
- RepoGraph demonstrates superior performance with extensive experiments (`cQhV`, `v4kH`, `wovV`, `btxE`).
- RepoGraph is transferable to repository-level tasks (`wovV`, `cQhV`, `btxE`).
- This paper provides insightful and thorough analyses (`cQhV`, `wovV`, `v4kH`).
- The paper is well-written (v4kH, wovV) with code for reproducibility (`H4KK`).

We’ve responded individually to each reviewer’s questions. To address your concerns and enhance our submission, we’ve incorporated your suggestions to conduct further experiments and provide additional results and analyses in the revised PDF. Below is a summary of the key updates:
- [Experiments with Claude-3.5-Sonnet on SWE-bench-Lite (Appendix C.1)](https://openreview.net/forum?id=dw9VUsSHGB&noteId=rrSlBj4Zna).
- [Experiments with Deepseek-Coder-V2 on CrossCodeEval (Section 5.3)](https://openreview.net/forum?id=dw9VUsSHGB&noteId=RPezI3ZOy6).
- [A clearer and more detailed explanation of the methodology section with additional figures (Section 3.1, Figure 10, Figure 11)](https://openreview.net/forum?id=dw9VUsSHGB&noteId=cx5eIJaFjN).
- [Quantitative analysis results for both pass and fail cases (Appendix C.2)](https://openreview.net/forum?id=dw9VUsSHGB&noteId=rrSlBj4Zna).
- [More comprehensive related work references (Section 2.2).](https://openreview.net/forum?id=dw9VUsSHGB&noteId=3mhB4uW8gp)
- Further proofreading and typo fixing.

This is a kind reminder that we are approaching the end of the discussion period. We sincerely hope that our responses have addressed the issues you’ve raised satisfactorily. We would be happy to discuss with you if there are any further points you wish us to clarify or address for a better score.

Sincerely,

Paper 8051 Authors

---

### Meta-Review · Area_Chair_jufp · 2024-12-14

**Metareview:**

This paper introduces a new framework that can provide guidance and serves as a repository-wide navigation for AI software engineers. A graph-learning-based algorithm is provided, where nodes correspond to the code-line level and edges correspond to connections of the nodes at the repo level. Extensive experiments are conducted to demonstrate that the framework can be applied to realistic repository-level code completion tasks.

The motivation of this work is clear with an easy-to-implement method. The presentation and organization are overall good. Experiments and following discussions are also comprehensive. The potential drawback of this work is that it is limited to Python and experiments can be enhanced by introducing human evolution studies. In general, this work makes valuable contributions to AI software engineering, and can inspire future work. Therefore, AC recommends accepting it.

**Additional Comments On Reviewer Discussion:**

This submission received useful comments from a total of five reviewers. The discussion and changes during the rebuttal period are summarized below.

- Reviewer cQhV commented that the experiments are not comprehensive. More convincing results should be introduced. Besides, some descriptions are confusing. During the rebuttal, the feedback addressed these concerns, which is acknowledged. The explanations and analysis should be reflected in the final version of this work.
- Reviewer H4KK mainly commented that the human evolution studies and related work can be supplemented. There are some unclear descriptions about Figure 3. The rebuttal addressed the concerns. Importantly, it is expected to include human evolution studies in the final version.
- Reviewer v4kH pointed out that the main weakness of this paper includes performance gains, cost assessment, and reproducibility. These are well answered during the rebuttal, with an increased rating.
- Reviewer wovV asked about the costs of each test and more experiments by involving DeepSeek and Qwen, which are handled during rebuttal.
- Reviewer btxE raised several concerns about unclear descriptions and algorithm details. After the rebuttal, most of the concerns are addressed. The remaining concern is that this work is somewhat limited to Python, and should be extended to other programming languages. Due to the limited rebuttal time, the authors provide some discussions. AC acknowledged its validity and the contribution of this paper, although the experiments are mainly conducted for Python. More results about the discussion can be provided in the camera-ready paper to enhance the content completeness of this work.

Based on the above summarization, this paper is considered to be above the acceptance and is recommended to be accepted.

---

### Decision · Program_Chairs · 2025-01-22

Accept (Poster)